# Neural Bellman-Ford Networks: A General Graph Neural Network Framework for Link Prediction

**Zhaocheng Zhu[1,2], Zuobai Zhang[1,2], Louis-Pascal Xhonneux[1,2], Jian Tang[1,3,4]**
Mila - Québec AI Institute[1], Université de Montréal[2]
HEC Montréal[3], CIFAR AI Chair[4]
{zhaocheng.zhu, zuobai.zhang, louis-pascal.xhonneux}@mila.quebec
jian.tang@hec.ca

## Abstract

Link prediction is a very fundamental task on graphs. Inspired by traditional path-based methods, in this paper we propose a general and flexible representation learning framework based on paths for link prediction. Specifically, we define the representation of a pair of nodes as the *generalized sum* of all path representations between the nodes, with each path representation as the *generalized product* of the edge representations in the path. Motivated by the Bellman-Ford algorithm for solving the shortest path problem, we show that the proposed path formulation can be efficiently solved by the generalized Bellman-Ford algorithm. To further improve the capacity of the path formulation, we propose the Neural Bellman-Ford Network (NBFNet), a general graph neural network framework that solves the path formulation with learned operators in the generalized Bellman-Ford algorithm. The NBFNet parameterizes the generalized Bellman-Ford algorithm with 3 neural components, namely INDICATOR, MESSAGE and AGGREGATE functions, which corresponds to the boundary condition, *multiplication* operator, and *summation* operator respectively[1]. The NBFNet covers many traditional path-based methods, and can be applied to both homogeneous graphs and multi-relational graphs (e.g., knowledge graphs) in both transductive and inductive settings. Experiments on both homogeneous graphs and knowledge graphs show that the proposed NBFNet outperforms existing methods by a large margin in both transductive and inductive settings, achieving new state-of-the-art results[2].

## 1 Introduction

Predicting the interactions between nodes (a.k.a. link prediction) is a fundamental task in the field of graph machine learning. Given the ubiquitous existence of graphs, such a task has many applications, such as recommender system [34], knowledge graph completion [41] and drug repurposing [27].

Traditional methods of link prediction usually define different heuristic metrics over the paths between a pair of nodes. For example, Katz index [30] is defined as a weighted count of paths between two nodes. Personalized PageRank [42] measures the similarity of two nodes as the random walk probability from one to the other. Graph distance [37] uses the length of the shortest path between two nodes to predict their association. These methods can be directly applied to new graphs, i.e., inductive setting, enjoy good interpretability and scale up to large graphs. However, they are designed based on handcrafted metrics and may not be optimal for link prediction on real-world graphs.

---

[1]Unless stated otherwise, we use *summation* and *multiplication* to refer the generalized operators in the path formulation, rather than the basic operations of arithmetic.

[2]Code is available at `https://github.com/DeepGraphLearning/NBFNet`

35th Conference on Neural Information Processing Systems (NeurIPS 2021).

To address these limitations, some link prediction methods adopt graph neural networks (GNNs) [32, 48, 59] to automatically extract important features from local neighborhoods for link prediction. Thanks to the high expressiveness of GNNs, these methods have shown state-of-the-art performance. However, these methods can only be applied to predict new links on the training graph, i.e. transductive setting, and lack interpretability. While some recent methods [73, 55] extract features from local subgraphs with GNNs and support inductive setting, the scalability of these methods is compromised.

Therefore, we wonder if there exists an approach that enjoys the advantages of both traditional path-based methods and recent approaches based on graph neural networks, i.e., **generalization in the inductive setting**, **interpretability**, **high model capacity** and **scalability**.

In this paper, we propose such a solution. Inspired by traditional path-based methods, our goal is to develop a general and flexible representation learning framework for link prediction based on the paths between two nodes. Specifically, we define the representation of a pair of nodes as the *generalized sum* of all the path representations between them, where each path representation is defined as the *generalized product* of the edge representations in the path. Many link prediction methods, such as Katz index [30], personalized PageRank [42], graph distance [37], as well as graph theory algorithms like widest path [4] and most reliable path [4], are special instances of this path formulation with different *summation* and *multiplication* operators. Motivated by the polynomial-time algorithm for the shortest path problem [5], we show that such a formulation can be efficiently solved via the generalized Bellman-Ford algorithm [4] under mild conditions and scale up to large graphs.

The operators in the generalized Bellman-Ford algorithm—*summation* and *multiplication*—are handcrafted, which have limited flexibility. Therefore, we further propose the Neural Bellman-Ford Networks (NBFNet), a graph neural network framework that solves the above path formulation with learned operators in the generalized Bellman-Ford algorithm. Specifically, NBFNet parameterizes the generalized Bellman-Ford algorithm with three neural components, namely INDICATOR, MESSAGE and AGGREGATE functions. The INDICATOR function initializes a representation on each node, which is taken as the boundary condition of the generalized Bellman-Ford algorithm. The MESSAGE and the AGGREGATE functions learn the *multiplication* and *summation* operators respectively.

We show that the MESSAGE function can be defined according to the relational operators in knowledge graph embeddings [6, 68, 58, 31, 52], e.g., as a translation in Euclidean space induced by the relational operators of TransE [6]. The AGGREGATE function can be defined as learnable set aggregation functions [71, 65, 9]. With such parameterization, NBFNet can generalize to the inductive setting, meanwhile achieve one of the lowest time complexity among inductive GNN methods. A comparison of NBFNet and other GNN frameworks for link prediction is showed in Table 1. With other instantiations of MESSAGE and AGGREGATE functions, our framework can also recover some existing works on learning logic rules [69, 46] for link prediction on knowledge graphs (Table 2).

Our NBFNet framework can be applied to several link prediction variants, covering not only single-relational graphs (e.g., homogeneous graphs) but also multi-relational graphs (e.g., knowledge graphs). We empirically evaluate the proposed NBFNet for link prediction on homogeneous graphs and knowledge graphs in both transductive and inductive settings. Experimental results show that the proposed NBFNet outperforms existing state-of-the-art methods by a large margin in all settings, with an average relative performance gain of 18% on knowledge graph completion (HITS@1) and 22% on inductive relation prediction (HITS@10). We also show that the proposed NBFNet is indeed interpretable by visualizing the top-k relevant paths for link prediction on knowledge graphs.

Table 1: Comparison of GNN frameworks for link prediction. The time complexity refers to the *amortized time* for predicting a single edge or triplet. $|\mathcal{V}|$ is the number of nodes, $|\mathcal{E}|$ is the number of edges, and $d$ is the dimension of representations. The wall time is measured on FB15k-237 test set with 40 CPU cores and 4 GPUs. We estimate the wall time of GraIL based on a downsampled test set.

| Method | Inductive[3] | Interpretable | Learned Representation | Time Complexity | Wall Time |
|---|---|---|---|---|---|
| VGAE [32] / RGCN [48] | | | ✓ | $O(d)$ | 18 secs |
| NeuralLP [69] / DRUM [46] | ✓ | ✓ | | $O\left(\frac{|\mathcal{E}|d}{|\mathcal{V}|} + d^2\right)$ | 2.1 mins |
| SEAL [73] / GraIL [55] | ✓ | | ✓ | $O(|\mathcal{E}|d^2)$ | ≈1 month |
| NBFNet | ✓ | ✓ | ✓ | $O\left(\frac{|\mathcal{E}|d}{|\mathcal{V}|} + d^2\right)$ | 4.0 mins |

## 2 Related Work

Existing work on link prediction can be generally classified into 3 main paradigms: path-based methods, embedding methods, and graph neural networks.

**Path-based Methods.** Early methods on homogeneous graphs compute the similarity between two nodes based on the weighted count of paths (Katz index [30]), random walk probability (personalized PageRank [42]) or the length of the shortest path (graph distance [37]). SimRank [28] uses advanced metrics such as the expected meeting distance on homogeneous graphs, which is extended by PathSim [51] to heterogeneous graphs. On knowledge graphs, Path Ranking [35, 15] directly uses relational paths as symbolic features for prediction. Rule mining methods, such as NeuralLP [69] and DRUM [46], learn probabilistic logical rules to weight different paths. Path representation methods, such as Path-RNN [40] and its successors [11, 62], encode each path with recurrent neural networks (RNNs), and aggregate paths for prediction. However, these methods need to traverse an exponential number of paths and are limited to very short paths, e.g., $\leq 3$ edges. To scale up path-based methods, All-Paths [57] proposes to efficiently aggregate all paths with dynamic programming. However, All-Paths is restricted to bilinear models and has limited model capacity. Another stream of works [64, 10, 22] learns an agent to collect useful paths for link prediction. While these methods can produce interpretable paths, they suffer from extremely sparse rewards and require careful engineering of the reward function [38] or the search strategy [50]. Some other works [8, 44] adopt variational inference to learn a path finder and a path reasoner for link prediction.

**Embedding Methods.** Embedding methods learn a distributed representation for each node and edge by preserving the edge structure of the graph. Representative methods include DeepWalk [43] and LINE [53] on homogeneous graphs, and TransE [6], DistMult [68] and RotatE [52] on knowledge graphs. Later works improve embedding methods with new score functions [58, 13, 31, 52, 54, 76] that capture common semantic patterns of the relations, or search the score function in a general design space [75]. Embedding methods achieve promising results on link prediction, and can be scaled to very large graphs using multiple GPUs [78]. However, embedding methods do not explicitly encode local subgraphs between node pairs and cannot be applied to the inductive setting.

**Graph Neural Networks.** Graph neural networks (GNNs) [47, 33, 60, 65] are a family of representation learning models that encode topological structures of graphs. For link prediction, the prevalent frameworks [32, 48, 12, 59] adopt an auto-encoder formulation, which uses GNNs to encode node representations, and decodes edges as a function over node pairs. Such frameworks are potentially inductive if the dataset provides node features, but are transductive only when node features are unavailable. Another stream of frameworks, such as SEAL [73] and GraIL [55], explicitly encodes the subgraph around each node pair for link prediction. While these frameworks are proved to be more powerful than the auto-encoder formulation [74] and can solve the inductive setting, they require to materialize a subgraph for each link, which is not scalable to large graphs. By contrast, our NBFNet explicitly captures the paths between two nodes for link prediction, meanwhile achieves a relatively low time complexity (Table 1). ID-GNN [70] formalizes link prediction as a conditional node classification task, and augments GNNs with the identity of the source node. While the architecture of NBFNet shares some spirits with ID-GNN, our model is motivated by the generalized Bellman-Ford algorithm and has theoretical connections with traditional path-based methods. There are also some works trying to scale up GNNs for link prediction by dynamically pruning the set of nodes in message passing [66, 20]. These methods are complementary to NBFNet, and may be incorporated into our method to further improve scalability.

## 3 Methodology

In this section, we first define a path formulation for link prediction. Our path formulation generalizes several traditional methods, and can be efficiently solved by the generalized Bellman-Ford algorithm. Then we propose Neural Bellman-Ford Networks to learn the path formulation with neural functions.

### 3.1 Path Formulation for Link Prediction

We consider the link prediction problem on both knowledge graphs and homogeneous graphs. A

---

[3]We consider the inductive setting where a model can generalize to entirely new graphs without node features.

knowledge graph is denoted by $\mathcal{G} = (\mathcal{V}, \mathcal{E}, \mathcal{R})$, where $\mathcal{V}$ and $\mathcal{E}$ represent the set of entities (nodes) and relations (edges) respectively, and $\mathcal{R}$ is the set of relation types. We use $\mathcal{N}(u)$ to denote the set of nodes connected to $u$, and $\mathcal{E}(u)$ to denote the set of edges ending with node $u$. A homogeneous graph $\mathcal{G} = (\mathcal{V}, \mathcal{E})$ can be viewed as a special case of knowledge graphs, with only one relation type for all edges. Throughout this paper, we use **bold** terms, $\boldsymbol{w}_q(e)$ or $\boldsymbol{h}_q(u, v)$, to denote vector representations, and *italic* terms, $w_e$ or $w_{uv}$, to denote scalars like the weight of edge $(u, v)$ in homogeneous graphs or triplet $(u, r, v)$ in knowledge graphs. Without loss of generality, we derive our method based on knowledge graphs, while our method can also be applied to homogeneous graphs.

**Path Formulation.** Link prediction is aimed at predicting the existence of a query relation $q$ between a head entity $u$ and a tail entity $v$. From a representation learning perspective, this requires to learn a pair representation $\boldsymbol{h}_q(u, v)$, which captures the local subgraph structure between $u$ and $v$ w.r.t. the query relation $q$. In traditional methods, such a local structure is encoded by counting different types of random walks from $u$ to $v$ [35, 15]. Inspired by this construction, we formulate the pair representation as a *generalized sum* of path representations between $u$ and $v$ with a commutative *summation* operator $\oplus$. Each path representation $\boldsymbol{h}_q(P)$ is defined as a *generalized product* of the edge representations in the path with the *multiplication* operator $\otimes$.

$$\boldsymbol{h}_q(u, v) = \boldsymbol{h}_q(P_1) \oplus \boldsymbol{h}_q(P_2) \oplus ... \oplus \boldsymbol{h}_q(P_{|\mathcal{P}_{uv}|})|_{P_i \in \mathcal{P}_{uv}} \triangleq \bigoplus_{P \in \mathcal{P}_{uv}} \boldsymbol{h}_q(P) \tag{1}$$

$$\boldsymbol{h}_q(P = (e_1, e_2, ..., e_{|P|})) = \boldsymbol{w}_q(e_1) \otimes \boldsymbol{w}_q(e_2) \otimes ... \otimes \boldsymbol{w}_q(e_{|P|}) \triangleq \bigotimes_{i=1}^{|P|} \boldsymbol{w}_q(e_i) \tag{2}$$

where $\mathcal{P}_{uv}$ denotes the set of paths from $u$ to $v$ and $\boldsymbol{w}_q(e_i)$ is the representation of edge $e_i$. Note the *multiplication* operator $\otimes$ is not required to be commutative (e.g., matrix multiplication), therefore we define $\bigotimes$ to compute the product following the exact order. Intuitively, the path formulation can be interpreted as a depth-first-search (DFS) algorithm, where one searches all possible paths from $u$ to $v$, computes their representations (Equation 2) and aggregates the results (Equation 1). Such a formulation is capable of modeling several traditional link prediction methods, as well as graph theory algorithms. Formally, Theorem 1-5 state the corresponding path formulations for 3 link prediction methods and 2 graph theory algorithms respectively. See Appendix A for proofs.

**Theorem 1** *Katz index is a path formulation with $\oplus = +$, $\otimes = \times$ and $\boldsymbol{w}_q(e) = \beta w_e$.*

**Theorem 2** *Personalized PageRank is a path formulation with $\oplus = +$, $\otimes = \times$ and $\boldsymbol{w}_q(e) = \alpha w_{uv} / \sum_{v' \in \mathcal{N}(u)} w_{uv'}$.*

**Theorem 3** *Graph distance is a path formulation with $\oplus = \min$, $\otimes = +$ and $\boldsymbol{w}_q(e) = w_e$.*

**Theorem 4** *Widest path is a path formulation with $\oplus = \max$, $\otimes = \min$ and $\boldsymbol{w}_q(e) = w_e$.*

**Theorem 5** *Most reliable path is a path formulation with $\oplus = \max$, $\otimes = \times$ and $\boldsymbol{w}_q(e) = w_e$.*

**Generalized Bellman-Ford Algorithm.** While the above formulation is able to model important heuristics for link prediction, it is computationally expensive since the number of paths grows exponentially with the path length. Previous works [40, 11, 62] that directly computes the exponential number of paths can only afford a maximal path length of 3. A more scalable solution is to use the generalized Bellman-Ford algorithm [4]. Specifically, assuming the operators $\langle \oplus, \otimes \rangle$ satisfy a semiring system [21] with *summation identity* $⓪_q$ and *multiplication identity* $①_q$, we have the following algorithm.

$$\boldsymbol{h}_q^{(0)}(u, v) \leftarrow \mathbb{1}_q(u = v) \tag{3}$$

$$\boldsymbol{h}_q^{(t)}(u, v) \leftarrow \left( \bigoplus_{(x, r, v) \in \mathcal{E}(v)} \boldsymbol{h}_q^{(t-1)}(u, x) \otimes \boldsymbol{w}_q(x, r, v) \right) \oplus \boldsymbol{h}_q^{(0)}(u, v) \tag{4}$$

where $\mathbb{1}_q(u = v)$ is the *indicator* function that outputs $①_q$ if $u = v$ and $⓪_q$ otherwise. $\boldsymbol{w}_q(x, r, v)$ is the representation for edge $e = (x, r, v)$ and $r$ is the relation type of the edge. Equation 3 is known as the boundary condition, while Equation 4 is known as the Bellman-Ford iteration. The high-level idea of the generalized Bellman-Ford algorithm is to **compute the pair representation $\boldsymbol{h}_q(u, v)$ for a given entity $u$, a given query relation $q$ and all $v \in \mathcal{V}$ in parallel**, and reduce the

total computation by the distributive property of *multiplication* over *summation*. Since $u$ and $q$ are fixed in the generalized Bellman-Ford algorithm, we may abbreviate $\boldsymbol{h}_q^{(t)}(u, v)$ as $\boldsymbol{h}_v^{(t)}$ when the context is clear. When $\oplus = min$ and $\otimes = +$, it recovers the original Bellman-Ford algorithm for the shortest path problem [5]. See Appendix B for preliminaries and the proof of the above algorithm.

**Theorem 6** *Katz index, personalized PageRank, graph distance, widest path and most reliable path can be solved via the generalized Bellman-Ford algorithm.*

Table 2: Comparison of operators in NBFNet and other methods from the view of path formulation.

| Class | Method | MESSAGE $\boldsymbol{w}_q(e_i) \otimes \boldsymbol{w}_q(e_j)$ | AGGREGATE $\boldsymbol{h}_q(P_i) \oplus \boldsymbol{h}_q(P_j)$ | INDICATOR $\mathbb{0}_q, \mathbb{1}_q$ | Edge Representation $\boldsymbol{w}_q(e)$ |
|---|---|---|---|---|---|
| Traditional Link Prediction | Katz Index [30] | $\boldsymbol{w}_q(e_i) \times \boldsymbol{w}_q(e_j)$ | $\boldsymbol{h}_q(P_i) + \boldsymbol{h}_q(P_j)$ | $0, 1$ | $\beta w_e$ |
| | Personalized PageRank [42] | $\boldsymbol{w}_q(e_i) \times \boldsymbol{w}_q(e_j)$ | $\boldsymbol{h}_q(P_i) + \boldsymbol{h}_q(P_j)$ | $0, 1$ | $\alpha w_{uv} / \sum_{v' \in \mathcal{N}(u)} w_{uv'}$ |
| | Graph Distance [37] | $\boldsymbol{w}_q(e_i) + \boldsymbol{w}_q(e_j)$ | $\min(\boldsymbol{h}_q(P_i), \boldsymbol{h}_q(P_j))$ | $+\infty, 0$ | $w_e$ |
| Graph Theory Algorithms | Widest Path [4] | $\min(\boldsymbol{w}_q(e_i), \boldsymbol{w}_q(e_j))$ | $\max(\boldsymbol{h}_q(P_i), \boldsymbol{h}_q(P_j))$ | $-\infty, +\infty$ | $w_e$ |
| | Most Reliable Path [4] | $\boldsymbol{w}_q(e_i) \times \boldsymbol{w}_q(e_j)$ | $\max(\boldsymbol{h}_q(P_i), \boldsymbol{h}_q(P_j))$ | $0, 1$ | $w_e$ |
| Logic Rules | NeuralLP [69] / DRUM [46] | $\boldsymbol{w}_q(e_i) \times \boldsymbol{w}_q(e_j)$ | $\boldsymbol{h}_q(P_i) + \boldsymbol{h}_q(P_j)$ | $0, 1$ | Weights learned by LSTM [23] |
| | NBFNet | Relational operators of knowledge graph embeddings [6, 68, 52] | Learned set aggregators [9] | Learned indicator functions | Learned relation embeddings |

## 3.2 Neural Bellman-Ford Networks

While the generalized Bellman-Ford algorithm can solve many classical methods (Theorem 6), these methods instantiate the path formulation with handcrafted operators (Table 2), and may not be optimal for link prediction. To improve the capacity of path formulation, we propose a general framework, Neural Bellman-Ford Networks (NBFNet), to learn the operators in the pair representations.

**Neural Parameterization.** We relax the semiring assumption and parameterize the generalized Bellman-Ford algorithm (Equation 3 and 4) with 3 neural functions, namely INDICATOR, MESSAGE and AGGREGATE functions. The INDICATOR function replaces the *indicator* function $\mathbb{1}_q(u = v)$. The MESSAGE function replaces the binary *multiplication* operator $\otimes$. The AGGREGATE function is a permutation invariant function over sets that replaces the n-ary *summation* operator $\bigoplus$. Note that one may alternatively define AGGREGATE as the commutative binary operator $\oplus$ and apply it to a sequence of messages. However, this will make the parameterization more complicated.

---

**Algorithm 1** Neural Bellman-Ford Networks

**Input:** source node $u$, query relation $q$, #layers $T$
**Output:** pair representations $\boldsymbol{h}_q(u, v)$ for all $v \in \mathcal{V}$

1: **for** $v \in \mathcal{V}$ **do** ▷ Boundary condition
2: $\quad \boldsymbol{h}_v^{(0)} \leftarrow$ INDICATOR$(u, v, q)$
3: **end for**
4: **for** $t \leftarrow 1$ to $T$ **do** ▷ Bellman-Ford iteration
5: $\quad$ **for** $v \in \mathcal{V}$ **do**
6: $\quad\quad \mathcal{M}_v^{(t)} \leftarrow \left\{ \boldsymbol{h}_v^{(0)} \right\}$ ▷ Message augmentation
7: $\quad\quad$ **for** $(x, r, v) \in \mathcal{E}(v)$ **do**
8: $\quad\quad\quad \boldsymbol{m}_{(x,r,v)}^{(t)} \leftarrow$ MESSAGE$^{(t)}(\boldsymbol{h}_x^{(t-1)}, \boldsymbol{w}_q(x, r, v))$
9: $\quad\quad\quad \mathcal{M}_v^{(t)} \leftarrow \mathcal{M}_v^{(t)} \cup \left\{ \boldsymbol{m}_{(x,r,v)}^{(t)} \right\}$
10: $\quad\quad$ **end for**
11: $\quad\quad \boldsymbol{h}_v^{(t)} \leftarrow$ AGGREGATE$^{(t)}(\mathcal{M}_v^{(t)})$
12: $\quad$ **end for**
13: **end for**
14: **return** $\boldsymbol{h}_v^{(T)}$ as $\boldsymbol{h}_q(u, v)$ for all $v \in \mathcal{V}$

---

Now consider the generalized Bellman-Ford algorithm for a given entity $u$ and relation $q$. In this context, we abbreviate $\boldsymbol{h}_q^{(t)}(u, v)$ as $\boldsymbol{h}_v^{(t)}$, i.e., a representation on entity $v$ in the $t$-th iteration. It should be stressed that $\boldsymbol{h}_v^{(t)}$ is still a pair representation, rather than a node representation. By substituting the neural functions into Equation 3 and 4, we get our Neural Bellman-Ford Networks.

$$\boldsymbol{h}_v^{(0)} \leftarrow \text{INDICATOR}(u, v, q) \tag{5}$$

$$\boldsymbol{h}_v^{(t)} \leftarrow \text{AGGREGATE}\left( \left\{ \text{MESSAGE}\left( \boldsymbol{h}_x^{(t-1)}, \boldsymbol{w}_q(x, r, v) \right) \middle| (x, r, v) \in \mathcal{E}(v) \right\} \cup \left\{ \boldsymbol{h}_v^{(0)} \right\} \right) \tag{6}$$

NBFNet can be interpreted as a novel GNN framework for learning pair representations. Compared to common GNN frameworks [32, 48] that compute the pair representation as two independent node representations $\boldsymbol{h}_q(u)$ and $\boldsymbol{h}_q(v)$, NBFNet initializes a representation on the source node $u$, and readouts the pair representation on the target node $v$. Intuitively, our framework can be viewed as a

source-specific message passing process, where every node learns a representation conditioned on the source node. The pseudo code of NBFNet is outlined in Algorithm 1.

**Design Space.** Now we discuss some principled designs for MESSAGE, AGGREGATE and INDICATOR functions by drawing insights from traditional methods. Note the potential design space for NBFNet is way larger than what is presented here, as one can always borrow MESSAGE and AGGREGATE from the arsenal of message-passing GNNs [19, 16, 60, 65].

For the MESSAGE function, traditional methods instantiate it as natural summation, natural multiplication or min over scalars. Therefore, we may use the vectorized version of summation or multiplication. Intuitively, summation of $h_x^{(t-1)}$ and $w_q(x, r, v)$ can be interpreted as a translation of $h_x^{(t-1)}$ by $w_q(x, r, v)$ in the pair representation space, while multiplication corresponds to scaling. Such transformations correspond to the relational operators [18, 45] in knowledge graph embeddings [6, 68, 58, 31, 52]. For example, translation and scaling are the relational operators used in TransE [6] and DistMult [68] respectively. We also consider the rotation operator in RotatE [52].

The AGGREGATE function is instantiated as natural summation, max or min in traditional methods, which are reminiscent of set aggregation functions [71, 65, 9] used in GNNs. Therefore, we specify the AGGREGATE function to be sum, mean, or max, followed by a linear transformation and a non-linear activation. We also consider the principal neighborhood aggregation (PNA) proposed in a recent work [9], which jointly learns the types and scales of the aggregation function.

The INDICATOR function is aimed at providing a non-trivial representation for the source node $u$ as the boundary condition. Therefore, we learn a query embedding $q$ for $①_q$ and define INDICATOR function as $\mathbb{1}(u = v) * q$. Note it is also possible to additionally learn an embedding for $⓪_q$. However, we find a single query embedding works better in practice.

The edge representations are instantiated as transition probabilities or length in traditional methods. We notice that an edge may have different contribution in answering different query relations. Therefore, we parameterize the edge representations as a linear function over the query relation, i.e., $w_q(x, r, v) = W_r q + b_r$. For homogeneous graphs or knowledge graphs with very few relations, we simplify the parameterization to $w_q(x, r, v) = b_r$ to prevent overfitting. Note that one may also parameterize $w_q(x, r, v)$ with learnable entity embeddings $x$ and $v$, but such a parameterization cannot solve the inductive setting. Similar to NeuralLP [69] & DRUM [46], we use different edge representations for different iterations, which is able to distinguish noncommutative edges in paths, e.g., father's mother v.s. mother's father.

**Link Prediction.** We now show how to apply the learned pair representations $h_q(u, v)$ to the link prediction problem. We predict the conditional likelihood of the tail entity $v$ as $p(v|u, q) = \sigma(f(h_q(u, v)))$, where $\sigma(\cdot)$ is the sigmoid function and $f(\cdot)$ is a feed-forward neural network. The conditional likelihood of the head entity $u$ can be predicted by $p(u|v, q^{-1}) = \sigma(f(h_{q^{-1}}(v, u)))$ with the same model. Following previous works [6, 52], we minimize the negative log-likelihood of positive and negative triplets (Equation 7). The negative samples are generated according to Partial Completeness Assumption (PCA) [14], which corrupts one of the entities in a positive triplet to create a negative sample. For undirected graphs, we symmetrize the representations and define $p_q(u, v) = \sigma(f(h_q(u, v) + h_q(v, u)))$. Equation 8 shows the loss for homogeneous graphs.

$$\mathcal{L}_{KG} = -\log p(u, q, v) - \sum_{i=1}^{n} \frac{1}{n} \log(1 - p(u_i', q, v_i')) \tag{7}$$

$$\mathcal{L}_{homo} = -\log p(u, v) - \sum_{i=1}^{n} \frac{1}{n} \log(1 - p(u_i', v_i')), \tag{8}$$

where $n$ is the number of negative samples per positive sample and $(u_i', q, v_i')$ and $(u_i', v_i')$ are the $i$-th negative samples for knowledge graphs and homogeneous graphs, respectively.

**Time Complexity.** One advantage of NBFNet is that it has a relatively low time complexity during inference[4]. Consider a scenario where a model is required to infer the conditional likelihood of all possible triplets $p(v|u, q)$. We group triplets with the same condition $u, q$ together, where each group contains $|\mathcal{V}|$ triplets. For each group, we only need to execute Algorithm 1 once to get their

---

[4]Although the same analysis can be applied to training on a fixed number of samples, we note it is less instructive since one can trade-off samples for performance, and the trade-off varies from method to method.

predictions. Since a small constant number of iterations $T$ is enough for NBFNet to converge (Table 6b), Algorithm 1 has a time complexity of $O(|\mathcal{E}|d + |\mathcal{V}|d^2)$, where $d$ is the dimension of representations. Therefore, the amortized time complexity for a single triplet is $O\left(\frac{|\mathcal{E}|d}{|\mathcal{V}|} + d^2\right)$. For a detailed derivation of time complexity of other GNN frameworks, please refer to Appendix C.

## 4 Experiment

### 4.1 Experiment Setup

We evaluate NBFNet in three settings, knowledge graph completion, homogeneous graph link prediction and inductive relation prediction. The former two are transductive settings, while the last is an inductive setting. For knowledge graphs, we use FB15k-237 [56] and WN18RR [13]. We use the standard transductive splits [56, 13] and inductive splits [55] of these datasets. For homogeneous graphs, we use Cora, Citeseer and PubMed [49]. Following previous works [32, 12], we split the edges into train/valid/test with a ratio of 85:5:10. Statistics of datasets can be found in Appendix E. Additional experiments of NBFNet on OGB [25] datasets can be found in Appendix G.

**Implementation Details.** Our implementation generally follows the open source codebases of knowledge graph completion[5] and homogeneous graph link prediction[6]. For knowledge graphs, we follow [69, 46] and augment each triplet $\langle u, q, v \rangle$ with a flipped triplet $\langle v, q^{-1}, u \rangle$. For homogeneous graphs, we follow [33, 32] and augment each node $u$ with a self loop $\langle u, u \rangle$. We instantiate NBFNet with 6 layers, each with 32 hidden units. The feed-forward network $f(\cdot)$ is set to a 2-layer MLP with 64 hidden units. ReLU is used as the activation function for all hidden layers. We drop out edges that directly connect query node pairs during training to encourage the model to capture longer paths and prevent overfitting. Our model is trained on 4 Tesla V100 GPUs for 20 epochs. We select the models based on their performance on the validation set. See Appendix F for more details.

**Evaluation.** We follow the filtered ranking protocol [6] for knowledge graph completion. For a test triplet $\langle u, q, v \rangle$, we rank it against all negative triplets $\langle u, q, v' \rangle$ or $\langle u', q, v \rangle$ that do not appear in the knowledge graph. We report mean rank (MR), mean reciprocal rank (MRR) and HITS at N (H@N) for knowledge graph completion. For inductive relation prediction, we follow [55] and draw 50 negative triplets for each positive triplet and use the above filtered ranking. We report HITS@10 for inductive relation prediction. For homogeneous graph link prediction, we follow [32] and compare the positive edges against the same number of negative edges. We report area under the receiver operating characteristic curve (AUROC) and average precision (AP) for homogeneous graphs.

**Baselines.** We compare NBFNet against path-based methods, embedding methods, and GNNs. These include 11 baselines for knowledge graph completion, 10 baselines for homogeneous graph link prediction and 4 baselines for inductive relation prediction. Note the inductive setting only includes path-based methods and GNNs, since existing embedding methods cannot handle this setting.

### 4.2 Main Results

Table 3 summarizes the results on knowledge graph completion. NBFNet significantly outperforms existing methods on all metrics and both datasets. NBFNet achieves an average relative gain of 21% in HITS@1 compared to the best path-based method, DRUM [46], on two datasets. Since DRUM is a special instance of NBFNet with natural summation and multiplication operators, this indicates the importance of learning MESSAGE and AGGREGATE functions in NBFNet. NBFNet also outperforms the best embedding method, LowFER [1], with an average relative performance gain of 18% in HITS@1 on two datasets. Meanwhile, NBFNet requires much less parameters than embedding methods. NBFNet only uses 3M parameters on FB15k-237, while TransE needs 30M parameters. See Appendix D for details on the number of parameters.

Table 4 shows the results on homogeneous graph link prediction. NBFNet gets the best results on Cora and PubMed, meanwhile achieves competitive results on CiteSeer. Note CiteSeer is extremely sparse (Appendix E), which makes it hard to learn good representations with NBFNet. One thing to note here is that unlike other GNN methods, NBFNet does not use the node features provided by

---

[5]`https://github.com/DeepGraphLearning/KnowledgeGraphEmbedding`. MIT license.
[6]`https://github.com/tkipf/gae`. MIT license.

Table 3: Knowledge graph completion results. Results of NeuraLP and DRUM are taken from [46]. Results of RotatE, HAKE and LowFER are taken from their original papers [52, 76, 1]. Results of the other embedding methods are taken from [52]. Since GraIL has scalability issues in this setting, we evaluate it with 50 and 100 negative triplets for FB15k-237 and WN18RR respectively and report MR based on an unbiased estimation.

| Class | Method | FB15k-237 | | | | | WN18RR | | | | |
|---|---|---|---|---|---|---|---|---|---|---|---|
| | | MR | MRR | H@1 | H@3 | H@10 | MR | MRR | H@1 | H@3 | H@10 |
| **Path-based** | Path Ranking [35] | 3521 | 0.174 | 0.119 | 0.186 | 0.285 | 22438 | 0.324 | 0.276 | 0.360 | 0.406 |
| | NeuralLP [69] | - | 0.240 | - | - | 0.362 | - | 0.435 | 0.371 | 0.434 | 0.566 |
| | DRUM [46] | - | 0.343 | 0.255 | 0.378 | 0.516 | - | 0.486 | 0.425 | 0.513 | 0.586 |
| **Embeddings** | TransE [6] | 357 | 0.294 | - | - | 0.465 | 3384 | 0.226 | - | - | 0.501 |
| | DistMult [68] | 254 | 0.241 | 0.155 | 0.263 | 0.419 | 5110 | 0.43 | 0.39 | 0.44 | 0.49 |
| | ComplEx [58] | 339 | 0.247 | 0.158 | 0.275 | 0.428 | 5261 | 0.44 | 0.41 | 0.46 | 0.51 |
| | RotatE [52] | 177 | 0.338 | 0.241 | 0.375 | 0.553 | 3340 | 0.476 | 0.428 | 0.492 | 0.571 |
| | HAKE [76] | - | 0.346 | 0.250 | 0.381 | 0.542 | - | 0.497 | 0.452 | 0.516 | 0.582 |
| | LowFER [1] | - | 0.359 | 0.266 | 0.396 | 0.544 | - | 0.465 | 0.434 | 0.479 | 0.526 |
| **GNNs** | RGCN [48] | 221 | 0.273 | 0.182 | 0.303 | 0.456 | 2719 | 0.402 | 0.345 | 0.437 | 0.494 |
| | GraIL [55] | 2053 | - | - | - | - | 2539 | - | - | - | - |
| | NBFNet | **114** | **0.415** | **0.321** | **0.454** | **0.599** | **636** | **0.551** | **0.497** | **0.573** | **0.666** |

Table 4: Homogeneous graph link prediction results. Results of VGAE and S-VGAE are taken from their original papers [32, 12].

| Class | Method | Cora | | Citeseer | | PubMed | |
|---|---|---|---|---|---|---|---|
| | | AUROC | AP | AUROC | AP | AUROC | AP |
| **Path-based** | Katz Index [30] | 0.834 | 0.889 | 0.768 | 0.810 | 0.757 | 0.856 |
| | Personalized PageRank [42] | 0.845 | 0.899 | 0.762 | 0.814 | 0.763 | 0.860 |
| | SimRank [28] | 0.838 | 0.888 | 0.755 | 0.805 | 0.743 | 0.829 |
| **Embeddings** | DeepWalk [43] | 0.831 | 0.850 | 0.805 | 0.836 | 0.844 | 0.841 |
| | LINE [53] | 0.844 | 0.876 | 0.791 | 0.826 | 0.849 | 0.888 |
| | node2vec [17] | 0.872 | 0.879 | 0.838 | 0.868 | 0.891 | 0.914 |
| **GNNs** | VGAE [32] | 0.914 | 0.926 | 0.908 | 0.920 | 0.944 | 0.947 |
| | S-VGAE [12] | 0.941 | 0.941 | **0.947** | **0.952** | 0.960 | 0.960 |
| | SEAL [73] | 0.933 | 0.942 | 0.905 | 0.924 | 0.978 | 0.979 |
| | TLC-GNN [67] | 0.934 | 0.931 | 0.909 | 0.916 | 0.970 | 0.968 |
| | NBFNet | **0.956** | **0.962** | 0.923 | 0.936 | **0.983** | **0.982** |

Table 5: Inductive relation prediction results (HITS@10). V1-v4 corresponds to the 4 standard versions of inductive splits. Results of compared methods are taken from [55].

| Class | Method | FB15k-237 | | | | WN18RR | | | |
|---|---|---|---|---|---|---|---|---|---|
| | | v1 | v2 | v3 | v4 | v1 | v2 | v3 | v4 |
| **Path-based** | NeuralLP [16] | 0.529 | 0.589 | 0.529 | 0.559 | 0.744 | 0.689 | 0.462 | 0.671 |
| | DRUM [46] | 0.529 | 0.587 | 0.529 | 0.559 | 0.744 | 0.689 | 0.462 | 0.671 |
| | RuleN [39] | 0.498 | 0.778 | 0.877 | 0.856 | 0.809 | 0.782 | 0.534 | 0.716 |
| **GNNs** | GraIL [55] | 0.642 | 0.818 | 0.828 | 0.893 | 0.825 | 0.787 | 0.584 | 0.734 |
| | NBFNet | **0.834** | **0.949** | **0.951** | **0.960** | **0.948** | **0.905** | **0.893** | **0.890** |

the datasets but is still able to outperform most other methods. We leave how to effectively combine node features and structural representations for link prediction as our future work.

Table 5 summarizes the results on inductive relation prediction. On all inductive splits of two datasets, NBFNet achieves the best result. NBFNet outperforms the previous best method, GraIL [55], with an average relative performance gain of 22% in HITS@10. Note that GraIL explicitly encodes the local subgraph surrounding each node pair and has a high time complexity (Appendix C). Usually, GraIL can at most encode a 2-hop subgraph, while our NBFNet can efficiently explore longer paths.

### 4.3 Ablation Study

**MESSAGE & AGGREGATE Functions.** Table 6a shows the results of different MESSAGE and AGGREGATE functions. Generally, NBFNet benefits from advanced embedding methods (DistMult,

RotatE > TransE) and aggregation functions (PNA > sum, mean, max). Among simple AGGREGATE functions (sum, mean, max), combinations of MESSAGE and AGGREGATE functions (TransE & max, DistMult & sum) that satisfy the semiring assumption[7] of the generalized Bellman-Ford algorithm, achieve locally optimal performance. PNA significantly improves over simple counterparts, which highlights the importance of learning more powerful AGGREGATE functions.

**Number of GNN Layers.** Table 6b compares the results of NBFNet with different number of layers. Although it has been reported that GNNs with deep layers often result in significant performance drop [36, 77], we observe NBFNet does not have this issue. The performance increases monotonically with more layers, hitting a saturation after 6 layers. We conjecture the reason is that longer paths have negligible contribution, and paths not longer than 6 are enough for link prediction.

**Performance by Relation Category.** We break down the performance of NBFNet by the categories of query relations: one-to-one, one-to-many, many-to-one and many-to-many[8]. Table 6c shows the prediction results for each category. It is observed that NBFNet not only improves on easy one-to-one cases, but also on hard cases where there are multiple true answers for the query.

Table 6: Ablation studies of NBFNet on FB15k-237. Due to space constraints, we only report MRR here. For full results on all metrics, please refer to Appendix H.

(a) Different MESSAGE and AGGREGATE functions.

| MESSAGE | AGGREGATE | | | |
|---|---|---|---|---|
| | Sum | Mean | Max | PNA [9] |
| TransE [6] | 0.297 | 0.310 | 0.377 | 0.383 |
| DistMult [69] | 0.388 | 0.384 | 0.374 | **0.415** |
| RotatE [52] | 0.392 | 0.376 | 0.385 | **0.414** |

(b) Different number of layers.

| Method | #Layers ($T$) | | | |
|---|---|---|---|---|
| | 2 | 4 | 6 | 8 |
| NBFNet | 0.345 | 0.409 | **0.415** | **0.416** |

(c) Performance w.r.t. relation category. The two scores are the rankings over heads and tails respectively.

| Method | Relation Category | | | |
|---|---|---|---|---|
| | 1-to-1 | 1-to-N | N-to-1 | N-to-N |
| TransE [6] | 0.498/0.488 | 0.455/0.071 | 0.079/0.744 | 0.224/0.330 |
| RotatE [51] | 0.487/0.484 | 0.467/0.070 | 0.081/0.747 | 0.234/0.338 |
| NBFNet | **0.578/0.600** | **0.499/0.122** | **0.165/0.790** | **0.348/0.456** |

## 4.4 Path Interpretations of Predictions

One advantage of NBFNet is that we can interpret its predictions through paths, which may be important for users to understand and debug the model. Intuitively, the interpretations should contain paths that contribute most to the prediction $p(u, q, v)$. Following local interpretation methods [3, 72], we approximate the local landscape of NBFNet with a linear model over the set of all paths, i.e., 1st-order Taylor polynomial. We define the importance of a path as its weight in the linear model, which can be computed by the partial derivative of the prediction w.r.t. the path. Formally, the top-k path interpretations for $p(u, q, v)$ are defined as

$$P_1, P_2, ..., P_k = \underset{P \in \mathcal{P}_{uv}}{\text{top-k}} \frac{\partial p(u, q, v)}{\partial P} \tag{9}$$

Note this formulation generalizes the definition of logical rules [69, 46] to non-linear models. While directly computing the importance of all paths is intractable, we approximate them with edge importance. Specifically, the importance of each path is approximated by the sum of the importance of edges in that path, where edge importance is obtained via auto differentiation. Then the top-k path interpretations are equivalent to the top-k longest paths on the edge importance graph, which can be solved by a Bellman-Ford-style beam search. Better approximation is left as a future work.

Table 7 visualizes path interpretations from FB15k-237 test set. While users may have different insights towards the visualization, here is our understanding. 1) In the first example, NBFNet learns

---

[7]Here semiring is discussed under the assumption of linear activation functions. Rigorously, no combination satisfies a semiring if we consider non-linearity in the model.

[8]The categories are defined same as [63]. We compute the average number of tails per head and the average number of heads per tail. The category is *one* if the average number is smaller than 1.5 and *many* otherwise.

soft logical entailment, such as *impersonate$^{-1}$* $\wedge$ *nationality* $\implies$ *nationality* and *ethnicity$^{-1}$* $\wedge$ *distribution* $\implies$ *nationality*. 2) In second example, NBFNet performs analogical reasoning by leveraging the fact that *Florence* is similar to *Rome*. 3) In the last example, NBFNet extracts longer paths, since there is no obvious connection between *Pearl Harbor (film)* and *Japanese language*.

Table 7: Path interpretations of predictions on FB15k-237 test set. For each query triplet, we visualize the top-2 path interpretations and their weights. Inverse relations are denoted with a superscript $^{-1}$.

| Query | $\langle u, q, v \rangle$: $\langle$*O. Hardy, nationality, U.S.*$\rangle$ |
|---|---|
| 0.243 | $\langle$*O. Hardy, impersonate$^{-1}$, R. Little*$\rangle$ $\wedge$ $\langle$*R. Little, nationality, U.S.*$\rangle$ |
| 0.224 | $\langle$*O. Hardy, ethnicity$^{-1}$, Scottish American*$\rangle$ $\wedge$ $\langle$*Scottish American, distribution, U.S.*$\rangle$ |
| **Query** | $\langle u, q, v \rangle$: $\langle$*Florence, vacationer, D.C. Henrie*$\rangle$ |
| 0.251 | $\langle$*Florence, contain$^{-1}$, Italy*$\rangle$ $\wedge$ $\langle$*Italy, capital, Rome*$\rangle$ $\wedge$ $\langle$*Rome, vacationer, D.C. Henrie*$\rangle$ |
| 0.183 | $\langle$*Florence, place live$^{-1}$, G.F. Handel*$\rangle$ $\wedge$ $\langle$*G.F. Handel, place live, Rome*$\rangle$ $\wedge$ $\langle$*Rome, vacationer, D.C. Henrie*$\rangle$ |
| **Query** | $\langle u, q, v \rangle$: $\langle$*Pearl Harbor (film), language, Japanese*$\rangle$ |
| 0.211 | $\langle$*Pearl Harbor (film), film actor, C.-H. Tagawa*$\rangle$ $\wedge$ $\langle$*C.-H. Tagawa, nationality, Japan*$\rangle$ $\wedge$ $\langle$*Japan, country of origin, Yu-Gi-Oh!*$\rangle$ $\wedge$ $\langle$*Yu-Gi-Oh!, language, Japanese*$\rangle$ |
| 0.208 | $\langle$*Pearl Harbor (film), film actor, C.-H. Tagawa*$\rangle$ $\wedge$ $\langle$*C.-H. Tagawa, nationality, Japan*$\rangle$ $\wedge$ $\langle$*Japan, official language, Japanese*$\rangle$ |

## 5 Discussion and Conclusion

**Limitations.** There are a few limitations for NBFNet. First, the assumption of the generalized Bellman-Ford algorithm requires the operators $\langle \oplus, \otimes \rangle$ to satisfy a semiring. Due to the non-linear activation functions in neural networks, this assumption does not hold for NBFNet, and we do not have a theoretical guarantee on the loss incurred by this relaxation. Second, NBFNet is only verified on simple edge prediction, while there are other link prediction variants, e.g., complex logical queries with conjunctions ($\wedge$) and disjunctions ($\vee$) [18, 45]. In the future, we would like to how NBFNet approximates the path formulation, as well as apply NBFNet to other link prediction settings.

**Social Impacts.** Link prediction has a wide range of beneficial applications, including recommender systems, knowledge graph completion and drug repurposing. However, there are also some potentially negative impacts. First, NBFNet may encode the bias present in the training data, which leads to stereotyped predictions when the prediction is applied to a user on a social or e-commerce platform. Second, some harmful network activities could be augmented by powerful link prediction models, e.g., spamming, phishing, and social engineering. We expect future studies will mitigate these issues.

**Conclusion.** We present a representation learning framework based on paths for link prediction. Our path formulation generalizes several traditional methods, and can be efficiently solved via the generalized Bellman-Ford algorithm. To improve the capacity of the path formulation, we propose NBFNet, which parameterizes the generalized Bellman-Ford algorithm with learned INDICATOR, MESSAGE, AGGREGATE functions. Experiments on knowledge graphs and homogeneous graphs show that NBFNet outperforms a wide range of methods in both transductive and inductive settings.

## Acknowledgements

We would like to thank Komal Teru for discussion on inductive relation prediction, Guyue Huang for discussion on fused message passing implementation, and Yao Lu for assistance on large-scale GPU training. We thank Meng Qu, Chence Shi and Minghao Xu for providing feedback on our manuscript.

This project is supported by the Natural Sciences and Engineering Research Council (NSERC) Discovery Grant, the Canada CIFAR AI Chair Program, collaboration grants between Microsoft Research and Mila, Samsung Electronics Co., Ltd., Amazon Faculty Research Award, Tencent AI Lab Rhino-Bird Gift Fund and a NRC Collaborative R&D Project (AI4D-CORE-06). This project was also partially funded by IVADO Fundamental Research Project grant PRF-2019-3583139727. The computation resource of this project is supported by Calcul Québec[9] and Compute Canada[10].

---

[9]https://www.calculquebec.ca/
[10]https://www.computecanada.ca/

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
