# A Path Formulations for Traditional Methods

Here we demonstrate our path formulation is capable of modeling traditional link prediction methods like Katz index [30], personalized PageRank [42] and graph distance [37], as well as graph theory algorithms like widest path [4] and most reliable path [4].

Recall the path formulation is defined as

$$\boldsymbol{h}_q(u,v) = \boldsymbol{h}_q(P_1) \oplus \boldsymbol{h}_q(P_2) \oplus ... \oplus \boldsymbol{h}_q(P_{|\mathcal{P}_{uv}|})|_{P_i \in \mathcal{P}_{uv}} \triangleq \bigoplus_{P \in \mathcal{P}_{uv}} \boldsymbol{h}_q(P) \tag{1}$$

$$\boldsymbol{h}_q(P = (e_1, e_2, ..., e_{|P|})) = \boldsymbol{w}_q(e_1) \otimes \boldsymbol{w}_q(e_2) \otimes ... \otimes \boldsymbol{w}_q(e_{|P|}) \triangleq \bigotimes_{i=1}^{|P|} \boldsymbol{w}_q(e_i) \tag{2}$$

which can be written in the following compact form

$$\boldsymbol{h}_q(u,v) = \bigoplus_{P \in \mathcal{P}_{uv}} \bigotimes_{i=1}^{|P|} \boldsymbol{w}_q(e_i) \tag{10}$$

## A.1 Katz Index

The Katz index for a pair of nodes $u$, $v$ is defined as a weighted count of paths between $u$ and $v$, penalized by an attenuation factor $\beta \in (0,1)$. Formally, it can be written as

$$\text{Katz}(u,v) = \sum_{t=1}^{\infty} \beta^t \boldsymbol{e}_u^\top \boldsymbol{A}^t \boldsymbol{e}_v \tag{11}$$

where $\boldsymbol{A}$ denotes the adjacency matrix and $\boldsymbol{e}_u$, $\boldsymbol{e}_v$ denote the one-hot vector for nodes $u$, $v$ respectively. The term $\boldsymbol{e}_u^\top \boldsymbol{A}^t \boldsymbol{e}_v$ counts all paths of length $t$ between $u$, and $v$ and shorter paths are assigned with larger weights.

**Theorem 1** *Katz index is a path formulation with $\oplus = +$, $\otimes = \times$ and $\boldsymbol{w}_q(e) = \beta w_e$.*

**Proof.** We show that $\text{Katz}(u,v)$ can be transformed into a summation over all paths between $u$ and $v$, where each path is represented by a product of damped edge weights in the path. Mathematically, it can be derived as

$$\text{Katz}(u,v) = \sum_{t=1}^{\infty} \beta^t \sum_{P \in \mathcal{P}_{uv}:|P|=t} \prod_{e \in P} w_e \tag{12}$$

$$= \sum_{P \in \mathcal{P}_{uv}} \prod_{e \in P} \beta w_e \tag{13}$$

Therefore, the Katz index can be viewed as a path formulation with the *summation* operator $+$, the *multiplication* operator $\times$ and the edge representations $\beta w_e$. $\square$

## A.2 Personalized PageRank

The personalized PageRank (PPR) for $u$ computes the stationary distribution over nodes generated by an infinite random walker, where the walker moves to a neighbor node with probability $\alpha$ and returns to the source node $u$ with probability $1 - \alpha$ at each step. The probability of a node $v$ from a source node $u$ has the following closed-form solution [29]

$$\text{PPR}(u,v) = (1-\alpha) \sum_{t=1}^{\infty} \alpha^t \boldsymbol{e}_u^\top (\boldsymbol{D}^{-1}\boldsymbol{A})^t \boldsymbol{e}_v \tag{14}$$

where $\boldsymbol{D}$ is the degree matrix and $\boldsymbol{D}^{-1}\boldsymbol{A}$ is the (random walk) normalized adjacency matrix. Note that $\boldsymbol{e}_u^\top (\boldsymbol{D}^{-1}\boldsymbol{A})^t \boldsymbol{e}_v$ computes the probability of $t$-step random walks from $u$ to $v$.

**Theorem 2** *Personalized PageRank is a path formulation with $\oplus = +$, $\otimes = \times$ and $\boldsymbol{w}_q(e) = \alpha w_{uv} / \sum_{v' \in \mathcal{N}(u)} w_{uv'}$.*

**Proof.** We omit the coefficient $1 - \alpha$, since it is always positive and has no effect on the ranking of different node pairs. Then we have

$$\text{PPR}(u, v) \propto \sum_{t=1}^{\infty} \alpha^t \sum_{P \in \mathcal{P}_{uv}: |P|=t} \prod_{(a,b) \in P} \frac{w_{ab}}{\sum_{b' \in \mathcal{N}(a)} w_{ab'}} \tag{15}$$

$$= \sum_{P \in \mathcal{P}_{uv}} \prod_{(a,b) \in P} \frac{\alpha w_{ab}}{\sum_{b' \in \mathcal{N}(a)} w_{ab'}} \tag{16}$$

where the *summation* operator is $+$, the *multiplication* operator is $\times$ and edge representations are random walk probabilities scaled by $\alpha$. $\quad\square$

### A.3  Graph Distance

Graph distance (GD) is defined as the minimum length of all paths between $u$ and $v$.

**Theorem 3** *Graph distance is a path formulation with* $\oplus = \min$, $\otimes = +$ *and* $\boldsymbol{w}_q(e) = w_e$.

**Proof.** Since the length of a path is the sum of edge lengths in the path, we have

$$\text{GD}(u, v) = \min_{P \in \mathcal{P}_{uv}} \sum_{e \in P} w_e \tag{17}$$

Here the *summation* operator is $\min$, the *multiplication* operator is $+$ and the edge representations are the lengths of edges. $\quad\square$

### A.4  Widest Path

The widest path (WP), also known as the maximum capacity path, is aimed at finding a path between two given nodes, such that the path maximizes the minimum edge weight in the path.

**Theorem 4** *Widest path is a path formulation with* $\oplus = \max$, $\otimes = \min$ *and* $\boldsymbol{w}_q(e) = w_e$.

**Proof.** Given two nodes $u$ and $v$, we can write the widest path as

$$\text{WP}(u, v) = \max_{P \in \mathcal{P}_{uv}} \min_{e \in P} w_e \tag{18}$$

Here the *summation* operator is $\max$, the *multiplication* operator is $\min$ and the edge representations are plain edge weights. $\quad\square$

### A.5  Most Reliable Path

For a graph with non-negative edge probabilities, the most reliable path (MRP) is the path with maximal probability from a start node to an end node. This is also known as Viterbi algorithm [61] used in the maximum a posterior (MAP) inference of hidden Markov models (HMM).

**Theorem 5** *Most reliable path is a path formulation with* $\oplus = \max$, $\otimes = \times$ *and* $\boldsymbol{w}_q(e) = w_e$.

**Proof.** For a start node $u$ and an end node $v$, the probaility of their most reliable path is

$$\text{MRP}(u, v) = \max_{P \in \mathcal{P}_{uv}} \prod_{e \in P} w_e \tag{19}$$

Here the *summation* operator is $\max$, the *multiplication* operator is $\times$ and the edge representations are edge probabilities. $\quad\square$

## B  Generalized Bellman-Ford Algorithm

First, we prove that the path formulation can be efficiently solved by the generalized Bellman-Ford algorithm when the operators $\langle \oplus, \otimes \rangle$ satisfy a semiring. Then, we show that traditional methods satisfy the semiring assumption and therefore can be solved by the generalized Bellman-Ford algorithm.

## B.1 Preliminaries on Semirings

Semirings are algebraic structures with two operators, *summation* $\oplus$ and *multiplication* $\otimes$, that share similar properties with the natural summation and the natural multiplication defined on integers. Specifically, $\oplus$ should be commutative, associative and have an identity element ⓪. $\otimes$ should be associative and have an identity element ①. Mathematically, the *summation* $\oplus$ satisfies

- **Commutative Property.** $a \oplus b = b \oplus a$
- **Associative Property.** $(a \oplus b) \oplus c = a \oplus (b \oplus c)$
- **Identity Element.** $a \oplus ⓪ = a$

The *multiplication* $\otimes$ satisfies

- **Associative Property.** $(a \otimes b) \otimes c = a \otimes (b \otimes c)$
- **Absorption Property.** $a \otimes ⓪ = ⓪ \otimes a = ⓪$
- **Identity Element.** $a \otimes ① = ① \otimes a = a$

Additionally, $\otimes$ should be distributive over $\oplus$.

- **Distributive Property (Left).** $a \otimes (b \oplus c) = (a \otimes b) \oplus (a \otimes c)$
- **Distributive Property (Right).** $(b \oplus c) \otimes a = (b \otimes a) \oplus (c \otimes a)$

Note semirings differ from natural arithmetic operators in two aspects. First, the *summation* operator $\oplus$ does not need to be invertible, e.g., min or max. Second, the *multiplication* operator $\otimes$ does not need to be commutative nor invertible, e.g., matrix multiplication.

## B.2 Generalized Bellman-Ford Algorithm for Path Formulation

Now we prove that the generalized Bellman-Ford algorithm computes the path formulation when the operators $\langle \oplus, \otimes \rangle$ satisfy a semiring. It should be stressed that the generalized Bellman-Ford algorithm for path problems has been proved in [4], and not a contribution of this paper. Here we apply the proof to our proposed path formulation.

The generalized Bellman-Ford algorithm computes the following iterations for all $v \in \mathcal{V}$

$$\boldsymbol{h}_q^{(0)}(u, v) \leftarrow \mathbb{1}_q(u = v) \tag{3}$$

$$\boldsymbol{h}_q^{(t)}(u, v) \leftarrow \left( \bigoplus_{(x,r,v) \in \mathcal{E}(v)} \boldsymbol{h}_q^{(t-1)}(u, x) \otimes \boldsymbol{w}_q(x, r, v) \right) \oplus \boldsymbol{h}_q^{(0)}(u, v) \tag{4}$$

**Lemma 1** *After $t$ Bellman-Ford iterations, the intermediate representation $\boldsymbol{h}_q^{(t)}(u, v)$ aggregates all path representations within a length of $t$ edges for all $v$. That is*

$$\boldsymbol{h}_q^{(t)}(u, v) = \bigoplus_{P \in \mathcal{P}_{uv}: |P| \leq t} \bigotimes_{i=1}^{|P|} \boldsymbol{w}_q(e_i) \tag{20}$$

**Proof.** We prove Lemma 1 by induction. For the base case $t = 0$, there is a single path of length $0$ from $u$ to itself and no path to other nodes. Due to the product definition of path representations, a path of length $0$ is equal to the *multiplication* identity $①_q$. Similarly, a summation of no path is equal to the *summation* identity $⓪_q$. Therefore, we have $\boldsymbol{h}_q^{(0)}(u, v) = \mathbb{1}_q(u = v) = \bigoplus_{P \in \mathcal{P}_{uv}: |P|=0} \bigotimes_{i=1}^{|P|} \boldsymbol{w}_q(e_i)$.

For the inductive case $t > 0$, we consider the second-to-last node $x$ in each path if the path has a length larger than $0$. To avoid overuse of brackets, we use the convention that $\otimes$ and $\bigotimes$ have a higher

priority than $\oplus$ and $\bigoplus$.

$$\boldsymbol{h}_q^{(t)}(u,v) = \left( \bigoplus_{(x,r,v)\in\mathcal{E}(v)} \boldsymbol{h}_q^{(t-1)}(u,x) \otimes \boldsymbol{w}_q(x,r,v) \right) \oplus \boldsymbol{h}_q^{(0)}(u,v) \tag{21}$$

$$= \left[ \bigoplus_{(x,r,v)\in\mathcal{E}(v)} \left( \bigoplus_{P\in\mathcal{P}_{ux}:|P|\leq t-1} \bigotimes_{i=1}^{|P|} \boldsymbol{w}_q(e_i) \right) \otimes \boldsymbol{w}_q(x,r,v) \right] \oplus \boldsymbol{h}_q^{(0)}(u,v) \tag{22}$$

$$= \left\{ \bigoplus_{(x,r,v)\in\mathcal{E}(v)} \left[ \bigoplus_{P\in\mathcal{P}_{ux}:|P|\leq t-1} \left( \bigotimes_{i=1}^{|P|} \boldsymbol{w}_q(e_i) \right) \otimes \boldsymbol{w}_q(x,r,v) \right] \right\} \oplus \boldsymbol{h}_q^{(0)}(u,v) \tag{23}$$

$$= \left( \bigoplus_{P\in\mathcal{P}_{uv}:1\leq|P|\leq t} \bigotimes_{i=1}^{|P|} \boldsymbol{w}_q(e_i) \right) \oplus \left( \bigoplus_{P\in\mathcal{P}_{uv}:|P|=0} \bigotimes_{i=1}^{|P|} \boldsymbol{w}_q(e_i) \right) \tag{24}$$

$$= \bigoplus_{P\in\mathcal{P}_{uv}:|P|\leq t} \bigotimes_{i=1}^{|P|} \boldsymbol{w}_q(e_i), \tag{25}$$

where Equation 22 substitutes the inductive assumption for $\boldsymbol{h}_q^{(t-1)}(u,x)$, Equation 23 uses the distributive property of $\otimes$ over $\oplus$. $\square$

By comparing Lemma 1 and Equation 10, we can see the intermediate representation converges to our path formulation $\lim_{t\to\infty} \boldsymbol{h}_q^{(t)}(u,v) = \boldsymbol{h}_q(u,v)$. More specifically, at most $|\mathcal{V}|$ iterations are required if we only consider simple paths, i.e., paths without repeating nodes. In practice, for link prediction we find it only takes a very small number of iterations (e.g., $T=6$) to converge, since long paths make negligible contribution to the task.

### B.3 Traditional Methods

**Theorem 6** *Katz index, personalized PageRank, graph distance, widest path and most reliable path can be solved via the generalized Bellman-Ford algorithm.*

**Proof.** Given that the generalized Bellman-Ford algorithm solves the path formulation when $\langle \oplus, \otimes \rangle$ satisfy a semiring, we only need to show that the operators of the path formulations for traditional methods satisfy semiring structures.

Katz index (Theorem 1) and personalized PageRank (Theorem 2) use the natural summation $+$ and the natural multiplication $\times$, which obviously satisfy a semiring.

Graph distance (Theorem 3) uses $\min$ for *summation* and $+$ for *multiplication*. The corresponding identities are $⓪ = +\infty$ and $①= 0$. It is obvious that $+$ satisfies the associative property and has identity element 0.

- **Commutative Property.** $\min(a,b) = \min(b,a)$
- **Associative Property.** $\min(\min(a,b),c) = \min(a,\min(b,c))$
- **Identity Element.** $\min(a,+\infty) = a$
- **Absorption Property.** $a + \infty = \infty + a = +\infty$
- **Distributive Property (Left).** $a + \min(b,c) = \min(a+b, a+c)$
- **Distributive Property (Right).** $\min(b,c) + a = \min(b+a, c+a)$

Widest path (Theorem 4) uses $\max$ for *summation* and $\min$ for *multiplication*. The corresponding identities are $⓪ = -\infty$ and $① = +\infty$. We have

- **Commutative Property.** $\max(a,b) = \max(b,a)$
- **Associative Property.** $\max(\max(a,b),c) = \max(a,\max(b,c))$
- **Identity Element.** $\max(a,-\infty) = a$
- **Associative Property.** $\min(\min(a,b),c) = \min(a,\min(b,c))$
- **Absorption Property.** $\min(a,-\infty) = \min(-\infty,a) = -\infty$
- **Identity Element.** $\min(a,+\infty) = \min(+\infty,a) = a$

- **Distributive Property (Left).** $\min(a, \max(b, c)) = \max(\min(a, b), \min(a, c))$
- **Distributive Property (Right).** $\min(\max(b, c), a) = \max(\min(b, a), \min(c, a))$

where the distributive property can be proved by enumerating all possible orders of $a$, $b$ and $c$.

Most reliable path (Theorem 5) uses $\max$ for *summation* and $\times$ for *multiplication*. The corresponding identities are $\textcircled{0} = 0$ and $\textcircled{1} = 1$, since all path representations are probabilities in $[0, 1]$. It is obvious that $\times$ satisfies the associative property, the absorption property and has identity element 0.

- **Commutative Property.** $\max(a, b) = \max(b, a)$
- **Associative Property.** $\max(\max(a, b), c) = \max(a, \max(b, c))$
- **Identity Element.** $\max(a, 0) = a$
- **Distributive Property (Left).** $a \times \max(b, c) = \max(a \times b, a \times c)$
- **Distributive Property (Right).** $\max(b, c) \times a = \max(b \times a, c \times a)$

where the identity element and the distributive property hold for non-negative elements. $\qquad\square$

# C   Time Complexity of GNN Frameworks

Here we prove the time complexity for NBFNet and other GNN frameworks.

## C.1   NBFNet

**Lemma 2** *The time complexity of one NBFNet run (Algorithm 1) is $O(T(|\mathcal{E}|d + |\mathcal{V}|d^2))$.*

**Proof.**   We break the time complexity by INDICATOR, MESSAGE and AGGREGATE functions.

INDICATOR is called $|\mathcal{V}|$ times, and a single call to INDICATOR takes $O(d)$ time. MESSAGE is called $T(|\mathcal{E}| + |\mathcal{V}|)$ times, and a single call to MESSAGE, i.e., a relation operator, takes $O(d)$ time. AGGREGATE is called $T|\mathcal{V}|$ times over a total of $T|\mathcal{E}|$ messages with $d$ dimensions. Each call to AGGREGATE additionally takes $O(d^2)$ time due to the linear transformations in the function.

Therefore, the total complexity is summed to $O(T(|\mathcal{E}|d + |\mathcal{V}|d^2))$. $\quad\square$

In practice, we find a small constant $T$ works well for link prediction, and Lemma 2 can be reduced to $O(|\mathcal{E}|d + |\mathcal{V}|d^2)$ time.

Now consider applying NBFNet to infer the likelihood of all possible triplets. Without loss of generality, assume we want to predict the tail entity for each head entity and relation $p(v|u, q)$. We group triplets with the same condition $u, q$ together, where each group contains $|\mathcal{V}|$ triplets. For triplets in a group, we only need to execute Algorithm 1 once to get their predictions. Therefore, the amortized time for a single triplet is $O\left(\frac{|\mathcal{E}|d}{|\mathcal{V}|} + d^2\right)$.

## C.2   VGAE / RGCN

RGCN is a message-passing GNN applied to multi-relational graphs, with the message function being a per-relation linear transformation. VGAE can be viewed as a special case of RGCN applied to single-relational graphs. The time complexity of RGCN is similar to Lemma 2, except that each call to the message function takes $O(d^2)$ time due to the linear transformation. Therefore, the total complexity is $O(T(|\mathcal{E}|d^2 + |\mathcal{V}|d^2))$, where $T$ refers to the number of layers in RGCN. Since $|\mathcal{V}| \leq |\mathcal{E}|$, the complexity is reduced to $O(T|\mathcal{E}|d^2)$[11]. In practice, $T$ is a small constant and we get $O(|\mathcal{E}|d^2)$ complexity.

While directly executing RGCN once for each triplet is costly, a smart way to apply RGCN for inference is to first compute all node representations, and then perform link prediction with the node representations. The first step runs RGCN once for $|\mathcal{V}|^2|\mathcal{R}|$ triplets, while the second step takes $O(d)$ time. Therefore, the amortized time for a single triplet is $O\left(\frac{|\mathcal{E}|d^2}{|\mathcal{V}|^2|\mathcal{R}|} + d\right)$. For large graphs and reasonable choices of $d$, we have $|\mathcal{E}|d \leq |\mathcal{V}|^2|\mathcal{R}|$ and the amortized time can be reduced to $O(d)$.

---

[11]By moving the linear transformations from the message function to the aggregation function, one can also get an implementation of RGCN with $O(T|\mathcal{V}||\mathcal{R}|d^2)$ time, which is better for dense graphs but worse for sparse graphs. For knowledge graph datasets used in this paper, the above $O(T|\mathcal{E}|d^2)$ implementation is better.

## C.3 NeuralLP / DRUM

DRUM can be viewed as a special case of NBFNet with MESSAGE being Hadamard product and AGGREGATE being natural summation. NeuralLP is a special case of DRUM where the dimension $d$ equals to 1. Since there is no linear transformation in their AGGREGATE functions, the amortized time complexity for the message passing part is $O\left(\frac{T|\mathcal{E}|d}{|\mathcal{V}|}\right)$. Both DRUM and NeuralLP additionally use an LSTM to learn the edge weights for each layer, which additionally costs $O(Td^2)$ time for $T$ layers. $T$ is small and can be ignored like in other methods. Therefore, the amortized time of two parts is summed to $O\left(\frac{|\mathcal{E}|d}{|\mathcal{V}|} + d^2\right)$.

## C.4 SEAL / GraIL

GraIL first extracts a local subgraph surrounding the link, and then applies RGCN to the local subgraph. SEAL can be viewed as a special case of GraIL applied to single-relational graphs. Therefore, their amortized time is the same as that of one RGCN run, which is $O(|\mathcal{E}|d^2)$.

Note that one may still run GraIL on large graphs by restricting the local subgraphs to be very small, e.g., within 1-hop neighborhood of the query entities. However, this will severely harm the performance of link prediction. Moreover, most real-world graphs are small-world networks, and a moderate radius can easily cover a non-trivial number of nodes and edges, which costs a lot of time for GraIL.

# D  Number of Parameters

Table 8: Number of parameters in NBFNet. The number of parameters only grows with the number of relations $|\mathcal{R}|$, rather than the number of nodes $|\mathcal{V}|$ or edges $|\mathcal{E}|$. For FB15k-237 augmented with flipped triplets, $|\mathcal{R}|$ is 474. Our best configuration uses $T = 6$, $d = 32$ and hidden dimension $m = 64$.

|  | #Parameter | |
|---|---|---|
|  | **Analytic Formula** | **FB15k-237** |
| INDICATOR | $\|\mathcal{R}\|d$ | 15,168 |
| MESSAGE | $T\|\mathcal{R}\|d(d+1)$ | 3,003,264 |
| AGGREGATE | $Td(13d+3)$ | 80,448 |
| $f(\cdot)$ | $m(2d+1)+m+1$ | 4,225 |
| Total |  | 3,103,105 |

One advantage of NBFNet is that it requires much less parameters than embedding methods. For example, on FB15k-237, NBFNet requires 3M parameters while TransE requires 30M parameters. Table 8 shows a break down of number of parameters in NBFNet. Generally, the number of parameters in NBFNet scales linearly w.r.t. the number of relations, regardless the number of entities in the graph, which makes NBFNet more parameter-efficient for large graphs.

# E  Statistics of Datasets

Dataset statistics of two transductive settings, i.e., knowledge graph completion and homogeneous graph link prediction, are summarized in Table 9 and 10. Dataset statistics of inductive relation prediction is summarized in Table 11.

We use the standard transductive splits [56, 13] and inductive splits [55] for knowledge graphs. For homogeneous graphs, we follow previous works [32, 12] and randomly split the edges into train/validation/test sets with a ratio of 85:5:10. All the homogeneous graphs used in this paper are undirected. Note that for inductive relation prediction, the original paper [55] actually uses a *transductive valid set* that shares the same set of fact triplets as the training set for hyperparameter tuning. The *inductive test set* contains entities, query triplets and fact triplets that never appear in the training set. The same data split is adopted in this paper for a fair comparison.

Table 9: Dataset statistics for knowledge graph completion.

| Dataset | #Entity | #Relation | #Train | #Triplet #Validation | #Test |
|---|---|---|---|---|---|
| FB15k-237 [56] | 14,541 | 237 | 272,115 | 17,535 | 20,466 |
| WN18RR [13] | 40,943 | 11 | 86,835 | 3,034 | 3,134 |

Table 10: Dataset statistics for homogeneous link prediction.

| Dataset | #Node | #Train | #Edge #Validation | #Test |
|---|---|---|---|---|
| Cora [49] | 2,708 | 4,614 | 271 | 544 |
| CiteSeer [49] | 3,327 | 4,022 | 236 | 474 |
| PubMed [49] | 19,717 | 37,687 | 2,216 | 4,435 |

Table 11: Dataset statistics for inductive relation prediction. Queries refer to the triplets that are used as training or test labels, while facts are the triplets used as training or test inputs. In the training sets, all queries are also provided as facts.

| Dataset | | #Relation | Train #Entity | #Query | #Fact | Validation #Entity | #Query | #Fact | Test #Entity | #Query | #Fact |
|---|---|---|---|---|---|---|---|---|---|---|---|
| FB15k-237 [55] | v1 | 180 | 1,594 | 4,245 | 4,245 | 1,594 | 489 | 4,245 | 1,093 | 205 | 1,993 |
| | v2 | 200 | 2,608 | 9,739 | 9,739 | 2,608 | 1,166 | 9,739 | 1,660 | 478 | 4,145 |
| | v3 | 215 | 3,668 | 17,986 | 17,986 | 3,668 | 2,194 | 17,986 | 2,501 | 865 | 7,406 |
| | v4 | 219 | 4,707 | 27,203 | 27,203 | 4,707 | 3,352 | 27,203 | 3,051 | 1,424 | 11,714 |
| WN18RR [55] | v1 | 9 | 2,746 | 5,410 | 5,410 | 2,746 | 630 | 5,410 | 922 | 188 | 1,618 |
| | v2 | 10 | 6,954 | 15,262 | 15,262 | 6,954 | 1,838 | 15,262 | 2,757 | 441 | 4,011 |
| | v3 | 11 | 12,078 | 25,901 | 25,901 | 12,078 | 3,097 | 25,901 | 5,084 | 605 | 6,327 |
| | v4 | 9 | 3,861 | 7,940 | 7,940 | 3,861 | 934 | 7,940 | 7,084 | 1,429 | 12,334 |

# F  Implementation Details

Table 12: Hyperparameter configurations of NBFNet on different datasets. Adv. temperature corresponds to the temperature in self-adversarial negative sampling [52]. Note for FB15k-237 and WN18RR, we use the same hyperparameters for their transductive and inductive settings. We find our model configuration is robust across all datasets, therefore we only tune the learning hyperparameters for each dataset. All the hyperparameters are chosen by the performance on the validation set.

| Hyperparameter | | FB15k-237 | WN18RR | Cora | CiteSeer | PubMed |
|---|---|---|---|---|---|---|
| **GNN** | #layer($T$) | 6 | 6 | 6 | 6 | 6 |
| | hidden dim. | 32 | 32 | 32 | 32 | 32 |
| **MLP** | #layer | 2 | 2 | 2 | 2 | 2 |
| | hidden dim. | 64 | 64 | 64 | 64 | 64 |
| **Batch** | #positive | 256 | 128 | 256 | 256 | 64 |
| | #negative/#positive($n$) | 32 | 32 | 1 | 1 | 1 |
| **Learning** | optimizer | Adam | Adam | Adam | Adam | Adam |
| | learning rate | 5e-3 | 5e-3 | 5e-3 | 5e-3 | 5e-3 |
| | #epoch | 20 | 20 | 20 | 20 | 20 |
| | adv. temperature | 0.5 | 1 | - | - | - |

Our implementation generally follows the open source codebases of knowledge graph completion[12] and homogeneous graph link prediction[13]. Table 12 lists the hyperparameter configurations for different datasets. Table 13 shows the wall time of training and inference on different datasets.

**Data Augmentation.** For knowledge graphs, we follow previous works [69, 46] and augment each triplet $\langle u, q, v \rangle$ with a flipped triplet $\langle v, q^{-1}, u \rangle$. For homogeneous graphs, we follow previous works [33, 32] and augment each node $u$ with a self loop $\langle u, u \rangle$.

**Architecture Details.** We apply Layer Normalization [2] and short cut connection to accelerate the training of NBFNet. Layer Normalization is applied after each AGGREGATE function. The feed-forward network $f(\cdot)$ is instantiated as a MLP. ReLU is used as the activation function for all hidden layers. For undirected graphs, we symmetrize the pair representation by taking the sum of $\boldsymbol{h}_q(u, v)$ and $\boldsymbol{h}_q(v, u)$.

---

[12]https://github.com/DeepGraphLearning/KnowledgeGraphEmbedding. MIT license.
[13]https://github.com/tkipf/gae. MIT license.

**Training Details.** We train NBFNet on 4 Tesla V100 GPUs with standard data parallelism. During training, we drop out edges that directly connect query node pairs to encourage the model to capture longer paths and prevent overfitting. We select the best checkpoint for each model based on its performance on the validation set. The selection criteria is MRR for knowledge graphs and AUROC for homogeneous graphs.

**Fused Message Passing.** To reduce memory footprint and better utilize GPU hardware, we follow the efficient implementation of GNNs [26] and implement customized PyTorch operators that combines MESSAGE and AGGREGATE functions into a single operation, without creating all messages explicitly. This reduces the memory complexity of NBFNet from $O(|\mathcal{E}|d)$ to $O(|\mathcal{V}|d)$.

Table 13: Wall time of NBFNet on different datasets and in different settings (Table 3, 4 and 5). For inductive setting, the total time over 4 split versions is reported.

| Wall Time | Transductive | | | | | Inductive | |
| --- | --- | --- | --- | --- | --- | --- | --- |
| | FB15k-237 | WN18RR | Cora | CiteSeer | PubMed | FB15k-237 | WN18RR |
| **Training** | 9.7 hrs | 4.4 hrs | 5.5 mins | 5.3 mins | 5.6 hrs | 23 mins | 41 mins |
| **Inference** | 4.0 mins | 2.4 mins | < 1 sec | < 1 sec | 25 secs | 6 secs | 20 secs |

# G   Experimental Results on OGB Datasets

To demonstrate the effectiveness of NBFNet on large-scale graphs, we additionally evaluate our method on two knowledge graph datasets from OGB [25], ogbl-biokg and WikiKG90M. We follow the standard evaluation protocol of OGB link property prediction, and compute the mean reciprocoal rank (MRR) of the true entity against 1,000 negative entities.

## G.1   Results on ogbl-biokg

Ogbl-biokg is a large biomedical knowledge graph that contains 93,773 entities, 51 relations and 5,088,434 triplets. We compare NBFNet with 6 embedding methods on this dataset. Note by the time of this work, only embedding methods are available for such large-scale datasets. Table 14 shows the results on ogbl-biokg. NBFNet achieves the best result compared to all methods reported on the official leaderboard[14] with much fewer parameters. Note the previous best model AutoSF is based on architecture search and requires more computation resource than NBFNet for training.

Table 14: Knowledge graph completion results on ogbl-biokg. Results of compared methods are taken from the OGB leaderboard.

| Class | Method | Test MRR | Validation MRR | #Params |
| --- | --- | --- | --- | --- |
| **Embeddings** | TransE [6] | 0.7452 | 0.7456 | 187,648,000 |
| | DistMult [68] | 0.8043 | 0.8055 | 187,648,000 |
| | ComplEx [58] | 0.8095 | 0.8105 | 187,648,000 |
| | RotatE [52] | 0.7989 | 0.7997 | 187,597,000 |
| | AutoSF [75] | 0.8309 | **0.8317** | 93,824,000 |
| | PairRE [7] | 0.8164 | 0.8172 | 187,750,000 |
| **GNNs** | NBFNet | **0.8317** | **0.8318** | 734,209 |

## G.2   Results on WikiKG90M

WikiKG90M is an extremely large dataset used in OGB large-scale challenge [24], hold at KDD Cup 2021. It is a general-purpose knowledge graph containing 87,143,637 entities, 1,315 relations and 504,220,369 triplets.

To apply NBFNet to such a large scale, we use a bidirectional breath-first-search (BFS) algorithm to sample a local subgraph for each query. Given a query, we generate a $k$-hop neighborhood for each

---

[14]https://ogb.stanford.edu/docs/leader_linkprop/#ogbl-biokg

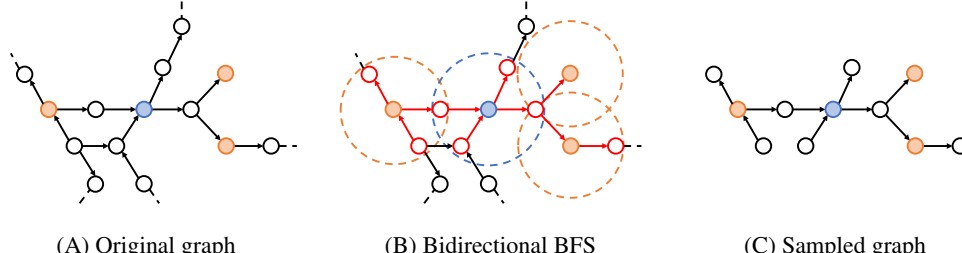

| (A) Original graph | (B) Bidirectional BFS | (C) Sampled graph |

Figure 1: Illustration of bidirectional BFS sampling. For a head entity and multiple tail candidates, we use BFS to sample a $k$-hop neighborhood around each entity, regardless of the direction of edges. The neighborhood is denoted by dashed circles. The nodes and edges visited by the BFS algorithm are extracted to generate the sampled graph. Best viewed in color.

of the head entity and the candidate tail entities, based on a BFS search. The union of all generated neighborhoods is then collected as the sampled graph. With this sampling algorithm, any path within a length of $2k$ between the head entity and any tail candidate is guaranteed to present in the sampled graph. See Figure 1 for illustration. While a standard single BFS algorithm computing the $2k$-hop neighborhood of the head entity has the same guarantee, a bidirectional BFS algorithm significantly reduces the number of nodes and edges in the sampled graph.

We additionally downsample the neighbors when expanding the neighbors of an entity, to tackle entities with large degrees. For each entity visited during the BFS algorithm, we downsample its outgoing neighbors and incoming neighbors to $m$ entities respectively.

Table 16 shows the results of NBFNet on WikiKG90M validation set. Our best single model uses $k = 2$ and $m = 100$. While the validation set requires to rank the true entity against 1,000 negative entities, in practice it is not mandatory to draw 1,000 negative samples for each positive sample during training. We find that reducing the negative samples from 1,000 to 20 and increasing the batch size from 4 to 64 provides a better result, although it creates a distribution shift between sampled graphs in training and validation. We leave further research of such distribution shift as a future work.

Table 15: Results of different MESSAGE and AGGREGATE functions on FB15k-237.

| MESSAGE | AGGREGATE | MR | MRR | H@1 | H@3 | H@10 |
|---|---|---|---|---|---|---|
| TransE [6] | Sum | 191 | 0.297 | 0.217 | 0.321 | 0.453 |
| | Mean | 161 | 0.310 | 0.218 | 0.339 | 0.496 |
| | Max | 135 | 0.377 | 0.282 | 0.415 | 0.565 |
| | PNA [9] | 129 | 0.383 | 0.288 | 0.420 | 0.568 |
| DistMult [68] | Sum | 136 | 0.388 | 0.294 | 0.427 | 0.574 |
| | Mean | 132 | 0.384 | 0.287 | 0.425 | 0.577 |
| | Max | 136 | 0.374 | 0.279 | 0.412 | 0.563 |
| | PNA [9] | **114** | **0.415** | **0.321** | **0.454** | **0.599** |
| RotatE [52] | Sum | 129 | 0.392 | 0.298 | 0.429 | 0.580 |
| | Mean | 138 | 0.376 | 0.278 | 0.416 | 0.571 |
| | Max | 139 | 0.385 | 0.290 | 0.423 | 0.572 |
| | PNA [9] | **117** | **0.414** | **0.323** | **0.454** | **0.593** |

Table 16: Knowledge graph completion results on WikiKG90M validation set.

| Model | Single Model | 6 Model Ensemble |
|---|---|---|
| **MRR** | 0.924 | 0.930 |

Table 17: Results of different number of layers on FB15k-237.

| #Layers ($T$) | MR | MRR | H@1 | H@3 | H@10 |
|---|---|---|---|---|---|
| 2 | 191 | 0.345 | 0.261 | 0.377 | 0.510 |
| 4 | 119 | 0.409 | 0.315 | 0.450 | 0.592 |
| 6 | **114** | **0.415** | **0.321** | **0.454** | **0.599** |
| 8 | 115 | 0.416 | 0.322 | 0.457 | 0.599 |

## H   Ablation Study

Table 15 shows the full results of different MESSAGE and AGGREGATE functions. Table 17 shows the full results of NBFNet with different number of layers.