# OpenReview forum: "Neural Bellman-Ford Networks: A General Graph Neural Network Framework for Link Prediction"
_NeurIPS.cc/2021/Conference — NeurIPS 2021 Poster_

### Official Review · Reviewer_HJVx · 2021-06-25

**Rating:** 7
**Confidence:** 5

**Summary:**

This paper considers a link prediction framework called Neural Bellman-Ford Networks. The idea is to encode the relational structure between two nodes by using GNN message passing. The key here is to put the identity of the source node in the input feature. The provided framework demonstrates SoTA performance on the link prediction and KG completion datasets.

**Ethical Concerns:**

No.

**Limitations And Societal Impact:**

Yes, may need to discuss more extensively on ID-GNN.

**Main Review:**

This is an interesting work in link prediction, and the performance is impressive. However, my biggest concern is the limited novelty, especially compared to the ID-GNN [1], which is not cited in the paper. ID-GNN already introduces the idea of using the node identity information in GNN's message passing and the idea of reformulating the link prediction as a conditional node prediction problem.

Hence, the main technical contributions of this paper are
(1) Show the theoretical connection to PPR, Katz index, etc (Table 1)
(2) Show more extensive experiments than ID-GNN.
(3) More ablation studies than ID-GNN.

I think the work still has some value. Especially, given that the experiments in ID-GNN are rather weak (mainly using synthetic graphs), this work serves as the empirical demonstration of the effectiveness of the ID-GNN framework in larger datasets.

One thing that would make this work more impactful from an empirical point of view is to try more established and standardized graph benchmarks such as the Open Graph Benchmark.  Both homogeneous and KG tasks are available, so it'd be great if authors can demonstrate the effectiveness there.

Overall, I enjoyed reading the paper, but it misses an important related work of [1]. I think this paper requires substantial reframing (not to over-claim the novelty, clarify the contribution given [1]) before being published at a conference.

[1] You et al. AAAI 2021 https://arxiv.org/abs/2101.10320

=============Post-rebuttal
The authors' response addressed my comments; Hence, I increased my score to 7. While I do believe careful reframing of the paper in light of the very relevant P-GNN is necessary, the big strength of the paper is in empirically demonstrating that a completely novel paradigm can outperform exiting KGE methods. Especially, it is impressive to see the methods can be even scaled up to the OGB-LSC graph.

**Time Spent Reviewing:**

1 hour

---

> ### Author Response · Authors · 2021-08-10
> **Response to Reviewer HJVx**
>
> Thanks for your comments and the related work you provided! Here is our response to your comments and concerns.
>
> **Q1: The key here is to put the identity of the source node in the input feature. ID-GNN already introduces the idea of using the node identity information in GNN's message passing and the idea of reformulating the link prediction as a conditional node prediction problem. Hence, the main technical contributions of this paper are (1) Show the theoretical connection to PPR, Katz index, etc (Table 1) (2) Show more extensive experiments than ID-GNN. (3) More ablation studies than ID-GNN.**
>
> A1: We agree that the instantiation of Indicator function and the conditional node prediction formulation in NBFNet may share some spirits with the identity coloring in ID-GNN, and therefore we will cite ID-GNN in our paper.
>
> However, NBFNet and ID-GNN are different in their tasks, motivations and techniques.
> 1) The tasks of NBFNet are knowledge graph completion, inductive relation prediction and link prediction, while the tasks of ID-GNN are node classification, link prediction and graph classification.
> 2) The motivation of NBFNet is to parameterize the generalized Bellman-Ford algorithm with neural networks to learn the path formulation, while the motivation of ID-GNN is to design GNNs with greater expressive power than 1-WL test.
> 3) Technically, NBFNet learns the Indicator function for the boundary condition, which is naturally derived from the generalized Bellman-Ford algorithm. By contrast, ID-GNN puts the identity information in the message function.
>
> **Q2: One thing that would make this work more impactful from an empirical point of view is to try more established and standardized graph benchmarks such as the Open Graph Benchmark. Both homogeneous and KG tasks are available, so it'd be great if authors can demonstrate the effectiveness there.**
>
> A2: Thanks for the suggestion. NBFNet was submitted to OGB-LSC and achieved 12th ranking among 28 teams in the competition. Please see the OGB-LSC WikiKG90M leaderboard[1] for results. Note results on the leaderboard heavily rely on ensemble techniques. Here we compare the validation result of our single model with the top 3 single models reported by the winner[2] and the runner up[3]. Our single model is better than the best single model reported by the winner and the runner up. NBFNet can easily scale to such large graphs with neighborhood sampling. We will illustrate the sampling techniques for OGB-LSC in our revision.
>
> | OGB-LSC | Valid MRR |
> |---------|-----------|
> | NOTE (winner's) | 0.9201 |
> | TransE (winner's) | 0.8800 |
> | RotatE (winner's) | 0.8940 |
> | TransE (runner up's) | 0.881 |
> | DistMult (runner up's) | 0.885 |
> | ComplEx (runner up's) | 0.885 |
> | NBFNet | **0.9237** |
>
> We also demonstrate the effectiveness of NBFNet on the link property prediction task of OGB. Here are our results on ogbl-biokg. NBFNet achieves the 2nd best result compared to all methods reported on the official leaderboard. Note the best model AutoSF is based on architecture search and requires more computation resource than NBFNet for training.
>
> | ogbl-biokg | Test MRR | Valid MRR | #Params |
> |------------|----------|-----------|---------|
> | TransE | 0.7452 | 0.7456 | 187,648,000 |
> | DistMult | 0.8043 | 0.8055 | 187,648,000 |
> | ComplEx | 0.8095 | 0.8105 | 187,648,000 |
> | RotatE | 0.7989 | 0.7997 | 187,597,000 |
> | PairRE | 0.8164 | 0.8172 | 187,750,000 |
> | AutoSF | **0.8309** | **0.8317** | *93,824,000* |
> | NBFNet | *0.8259* | *0.8281* | **734,209** |
>
> [1] OGB-LSC WikiKG90 leaderboard. [https://ogb.stanford.edu/kddcup2021/results/#final_wikikg90m](https://ogb.stanford.edu/kddcup2021/results/#final_wikikg90m)
>
> [2] Weiyue Su, et al. "NOTE: Solution for KDD-CUP 2021 WikiKG90M-LSC." arXiv preprint arXiv:2107.01892 (2021).
>
> [3] Weihua Peng, et al. "TECHNICAL REPORT OF WIKIKG90M-LSC." OGB KDDCup 2021.

---

> ### Author Response · Authors · 2021-08-15
> **Further suggestions are welcome**
>
> Thanks for your appreciation of our contribution. We will add a paragraph discussing GNNs with inductive information, such as P-GNN and ID-GNN, in the related works. We will also provide a section (or probably in the appendix if the space doesn't permit) to discuss the sampling techniques and experimental results for NBFNet on OGB-LSC. Feel free to reach out if you have further questions regarding the paper.

---

### Official Review · Reviewer_QoNE · 2021-07-08

**Rating:** 6
**Confidence:** 4

**Summary:**

The paper proposes an adaptive version of the generalised Bellman-Ford algorithm to solve link prediction problems in both knowledge bases and inductive settings. The resulting formulation is a slight variation of a standard graph neural network that accomodates different types of relations and computes pairwise node representations, that is, a different representation of node v for each neighbour.

**Limitations And Societal Impact:**

Yes.

**Main Review:**

The paper undoubtedly has merits, such as:

- Mentioning the similarities between the usual message passing formulation of GNNs and the aforementioned algorithm.
- Conducting an experimental comparison that looks fair (though something must be clarified by the authors), obtaining results that consistently surpass previous works. Despite this consistency ironically raising a red flag at the beginning, to the best of my knowledge the results seem solid. This may be attributed to the more general formulation (w.r.t. link prediction) of the proposed approach. Unfortunately, the code to inspect all the model selection and assessment phases is not available.

With that being said, the general feeling after reading the article is that it may have been written in a rush. This is due to the organisational choices and technical definitions that make it difficult to follow the discussion and are confusing in more than one passage. Moreover, some of the crucial claims of the authors about scalability and model capacity are mostly described in the appendix. Generally speaking, this may be perceived as unfair practice: appendices should support the main claims rather than address them. In other words, without reading the appendix, a reviewer would be unable to comprehensively evaluate all the claims made by these authors.

As concerns interpretability (the term is - wrongly - used intercheangebly with "explainability" by the authors), one of the fundamental challenges to be addressed by the authors, there is no solid qualitative evidence that the model, quote, "learns soft logical entailment" or "performs analogical reasoning". The three examples of Table 6 do not appear to be that meaningful. While we can argue about the most effective ways to "prove" interpretability, the evidence proposed by the authors cannot be considered sufficient.

Overall, it appears that the authors made too many claims and were unable to properly fit them into a single conference paper. While the idea and results are (again) noteworthy, the quality of the paper definitely suffers because of its presentation, and this has to be taken into consideration given the level of the conference. Improving the manuscript in this sense may require substantial rewritring that might not be feasible at this point.

Technical comments:

Abstract:

- please specify that the generalized sum is over the paths *between the nodes*.
- "very general" may be omitted

Section 1:

- Table 1 contains symbols that have not yet been introduced. Consider moving it at the end of Section 3. Also, a practical example of the MESSAGE, AGGREGATE and INDICATOR functions used in this work may help the reader.

Section 2:

- Could the authors please clarify why, on line 107, it is stated that autoencoders based on GNN do not work on the inductive setting? GNNs work on this setting off-the-shelf, so it is unclear to me why they should not generalise to new graphs at inference time. The authors reiterate this concept throughout the manuscript, including in Table 7 when R-GCN is defined as non-inductive, but it would be helpful to be on the same page about this.

Section 3:

- The mathematical formalisation is broken in more than one way, which makes the discussion confusing in many ways.
  = The use of the query symbol q is ambiguous. Is it on of the possible relations in R, or is it a triple (see table 6)?
  = The same is true for the definition og an edge: is it a tuple (u,v) or a triple (u,r,v)?
  = Similarly, w(e) != w(u,r,v) according to the definitions in the paper
  = Eqs 7 and 8: I believe the use of the expectation symbol in place of the mean is inappropriate. There are no formal distribution or random variables specified here. Please consider replacing the occurrences of the expectation with a uniformly weighted summation over the triples.
- Possible typo in Eq 4: should h_q^{(0)}  be should h_q^{(t-1)}?

- A suggestion: the abbreviation used before Eq 4 will confuse a lot of readers. The authors may consider expanding the notation despite its verbosity.

- What do the authors mean by "relatively low time complexity"? Is NBFNet truly scalable or not? It seems not, given the inference time comparison of Table C (which should have been included in the main article). Moreover, is it also scalable at training time? 17 hour for 20 epochs on PubMed seems a very slow training time.

- In general, it is not true that homogeneous graphs are undirected. Please rephrase.

Section 4:

- Why AllPath is not included in the empirical comparison? The authors stated it is the most similar solution compared to NBFNet, so we are expecting at least a comparison in the tables.
- Also, it would be nice to include a simple inductive baseline in Table 3 that applies a simple transformation to the node features before computing a link prediction score. On a personal note, I found out this strategy obtains very similar results to the ones reported by the authors.

- Table 5: did the authors fix the architecture, that is, 6 layers, based on the test results of Table 5? This would be a grave mistake (that would partially explain the consistent performance improvements over other methods). Please provide an honest answer.

- Could the authors please detail how Eq 9 generalizes logical rules to non-linear models? It seems quite a statement, but there is no enough evidence to support this.

Bibliography:

Please consider better referencing the literature rather than citing directly most papers as arxiv preprints. As the authors know, other authors benefit from having their work indexed as part of a conference rather than as a preprint. Here, there are pretty obvious examples of arxiv-referenced papers that have already been published at important conferences.


**Time Spent Reviewing:**

4

---

> ### Author Response · Authors · 2021-08-10
> **Response to Reviewer QoNE**
>
> Thanks for your constructive comments and suggestions! Here is our response to your concerns.
>
> **Q1: Could the authors please clarify why, on line 107, it is stated that autoencoders based on GNN do not work on the inductive setting? GNNs work on this setting off-the-shelf, so it is unclear to me why they should not generalise to new graphs at inference time.**
>
> A1: GNNs (e.g. GraphSAGE[1] and RGCN) can be applied to the inductive setting if the dataset provides input features for every entity. However, many knowledge graphs are not accompanied with node features, e.g. FB15k-237 & WN18RR used in our paper. In our paper, we follow the inductive setting introduced in GraIL[2], and require the model to generalize to entirely new graphs without node features. Autoencoder-based GNNs need to learn the embedding for each entity as model parameters, and can't be applied to this inductive setting. There is a detailed discussion of models available for the inductive setting in the related work section of [2], and we have already covered all these models in our experiments.
>
> [1] William L. Hamilton et al. "Inductive representation learning on large graphs." NIPS 2017.
>
> [2] Komal Teru, Etienne Denis, and Will Hamilton. "Inductive relation prediction by subgraph reasoning." ICML 2020.
>
> **Q2: It would be nice to include a simple inductive baseline in Table 3 that applies a simple transformation to the node features before computing a link prediction score.**
>
> A2: A simple inductive baseline can only be applied to graphs with input node features, but the knowledge graph datasets used in our paper don't contain node features.
>
> **Q3: What do the authors mean by "relatively low time complexity"? Is NBFNet truly scalable or not? It seems not, given the inference time comparison of Table C. Moreover, is it also scalable at training time? 17 hour for 20 epochs on PubMed seems a very slow training time.**
>
> A3: Relatively low time complexity is discussed w.r.t. other inductive methods (GraIL, NeuralLP, DRUM). NBFNet gets one of the lowest time complexity in inductive methods. We will clarify this in the revision.
>
> NBFNet is scalable at training time for knowledge graphs, but not for homogeneous graphs like PubMed. In knowledge graphs, negative samples are generated by corrupting one entity in a positive sample, which means we can group a positive and its negative samples as a single-source problem and execute NBFNet once. However, in homogeneous graphs, negative samples are randomly drawn from the missing links, and it is likely that positive and negative samples form a multi-source problem, which requires multiple runs of NBFNet.
>
> **Q4: Some of the crucial claims of the authors about scalability and model capacity are mostly described in the appendix. Generally speaking, this may be perceived as unfair practice: appendices should support the main claims rather than address them.**
>
> A4: We agree that claims should be supported in the main article. We conjecture the reason for you to perceive that scalability and high model capacity are ranked first in the contributions. Since scalability and model capacity are weaker contributions compared to the transductive & inductive performance and interpretability, we will reorder the contributions and modify the introduction accordingly to improve the organization of the paper.
>
> **Q5: As concerns interpretability (the term is - wrongly - used interchangeably with "explainability" by the authors), one of the fundamental challenges to be addressed by the authors, there is no solid qualitative evidence that the model, quote, "learns soft logical entailment" or "performs analogical reasoning". The three examples of Table 6 do not appear to be that meaningful.**
>
> A5: We think our method belongs to interpretability, and we will change our wording according to your suggestion. Here the contribution is that NBFNet can interpret its prediction by visualizing the top-weighted paths for the prediction. Note many previous SToA methods (embedding methods, GNNs) are not able to provide interpretations. It is not our goal to claim what logic NBFNet learns for prediction, and we will clarify this and weaken the descriptions in the revision.
>
> **Q6: Could the authors please detail how Eq 9 generalizes logical rules to non-linear models?**
>
> A6: For any non-linear model that is differentiable w.r.t. paths, its prediction can always be **locally approximated** by a linear model over paths. Since logical rules are defined as the top-weighted paths (or path types) from a linear model (NeuralLP, DRUM, RNNLogic[3]), we can generalize logical rules to a non-linear model by defining them over a local linear approximation of the non-linear model. The optimal local linear approximation is given by 1st Taylor polynomial, and its weights are equal to the partial derivatives in Equation 9. Therefore, Equation 9 computes the local logical rules for non-linear models.
>
> [3] Meng Qu, et al. "RNNLogic: Learning Logic Rules for Reasoning on Knowledge Graphs." ICLR 2020.
>
> **Q7: Table 5: did the authors fix the architecture, that is, 6 layers, based on the test results of Table 5?**
>
> A7: No. We always select the hyperparameters based on the performance on validation set. The weirdness in Table 5(b) is caused by our hardware constraints during the experiments. Hyperparameters for our main experiments are searched with 16GB GPUs, where 6 layers is the maximal possible value given the memory capacity. 8 layers is later carried out on 32GB GPUs. We will unify the experiments under the same hardware configuration in the revision.
>
> A partial evidence to prove the correctness of our model selection is that the same model checkpoint performs slightly better on the validation set than on the test set.
>
> |NBFNet|MR|MRR|HITS@1|HITS@3|HITS@10|
> |-|-|-|-|-|-|
> |FB15k-237valid|**135**|**0.382**|**0.288**|**0.421**|**0.567**|
> |FB15k-237test|139|0.378|0.285|0.417|0.563|
>
> **Q8: Why AllPaths is not included in the empirical comparison? The authors stated it is the most similar solution compared to NBFNet, so we are expecting at least a comparison in the tables.**
>
> A8: AllPaths didn't have results on the datasets used in our paper. Besides, its code isn't released. We try our best to reproduce AllPaths based on its equations. Here is the comparison of AllPaths and NBFNet on WN18RR in the transductive setting. Note AllPaths isn't inductive. AllPaths doesn't perform very well compared to NBFNet. We conjecture the reasons are: 1) AllPaths is a linear model developed in 2016, while NBFNet is a non-linear model with better parameterization inspired from embedding methods. 2) AllPaths shares the relation parameters for different time steps, which makes it very hard to train.
>
> |WN18RR|MR|MRR|HITS@1|HITS@3|HITS@10|
> |-|-|-|-|-|-|
> |AllPaths|20948|0.278|0.221|0.312|0.379|
> |NBFNet|**653**|**0.547**|**0.490**|**0.573**|**0.661**|
>
> **Q9: Possible typo in Eq 4: should h_q^{(0)} be should h_q^{(t-1)}?**
>
> A9: It is not a typo. A high-level intuition is that for paths between node $u$ and $v$ with length $0\leq|P|\leq t$, we can decompose them into paths with length $1\leq|P|\leq t$ and $0$-hop paths. Paths with length $1\leq |P|\leq t$ can be recursively solved via paths with length $0\leq|P'|\leq t-1$ plus 1 hop, while the $0$-hop paths are equal to $h_q^{(0)}(u, v)$. Please refer to Equation 21-25 for detailed derivations.
>
> **Q10: The use of the query symbol q is ambiguous. Is it one of the possible relations in R, or is it a triple (see table 6)? For the definition on an edge: is it a tuple (u,v) or a triple (u,r,v)? Similarly, w(e) != w(u,r,v) according to the definitions in the paper. Eqs 7 and 8: I believe the use of the expectation symbol in place of the mean is inappropriate. Please consider replacing the occurrences of the expectation with a uniformly weighted summation over the triples.**
>
> A10: $q$ is a relation in $\mathcal{R}$. We will change the notation in Table 6 to $\langle u,q,v \rangle=\langle O.~Hardy, nationality, U.S.\rangle$ to reduce ambiguity.
>
> The definition on an edge is a triplet $\langle u,r,v\rangle$ for knowledge graphs and a tuple $\langle u,v\rangle$ for homogeneous graphs. We will try our best to unify the notations under the umbrella of knowledge graphs, but in the context of homogeneous graphs, using $\langle u,v\rangle$ may be more intuitive for the readers.
>
> $w(e)$ refers to the representation of edge $e$, without emphasizing the edge is in knowledge graphs or homogeneous graphs. In the case of knowledge graphs, $w(e)$ refers to $w(u,r,v)$. In the case of homogeneous graphs, $w(e)$ refers to $w(u,v)$. We will clarify this in the revision.
>
> We agree to change the expectation to a uniformly weighted summation.
>
> **Q11: A suggestion: the abbreviation used before Eq 4 will confuse a lot of readers. The authors may consider expanding the notation despite its verbosity.**
>
> A11: Do you mean the abbreviation of $h_q^{(t)}(u,v)$ before Equation 5 and 6? We feel readers may mistake $|\mathcal{V}|$ hidden states for $|\mathcal{V}|^2$ hidden states if we use $h_q^{(t)}(u,v)$ in Equation 4. The abbreviation is a convention in single-source shortest path problem in computer science. A better writing could be first emphasizing that Bellman-Ford algorithm is designed for single-source problems in Equation 3 and 4, and then applying this abbreviation in generalized Bellman-Ford algorithm, rather than in NBFNet. We will modify this in the revision.
>
> **Q12: Abstract: Please specify that the generalized sum is over the paths *between the nodes*. "very general" may be omitted. Table 1 contains symbols that have not yet been introduced. Consider moving it at the end of Section 3. In general, it is not true that homogeneous graphs are undirected. Please rephrase. Please consider better referencing the literature rather than citing directly most papers as arxiv preprints.**
>
> A12: We agree with these suggestions. We will change them in the revision.

---

> > ### Comment · Reviewer_QoNE · 2021-08-19
> > **Response**
> >
> > Thanks for your response. It seems that the other reviewers did not feel the same about the presentation and organization of the paper. Since this is pretty subjective, I am willing to change my score.

---

> ### Author Response · Authors · 2021-08-15
> **Further suggestions are welcome**
>
> Thanks for your careful and detailed suggestions. We notice that many of your concerns (e.g. inductive, time complexity and interpretability) are crucial to determining the contribution and the position of this paper, and we wonder if you are satisfied with our feedback. If you believe these concerns still exist, we are open to more discussions and refinement.

---

### Official Review · Reviewer_4kGi · 2021-07-14

**Rating:** 6
**Confidence:** 4

**Summary:**

1. This paper proposes a new method for Link Prediction dubbed NBFNet, which is a hybrid approach that combines path based and GNN based approaches for Link Prediction.
2. The core idea of the paper is about computing the representation of a pair of nodes, which is used for predicting links between the nodes. This representation is computed using three neural operators namely INDICATOR, MESSAGE and AGGREGATE, which represent the parameterised version of Bellman-Ford algorithm.
3. Authors present experimental results on the tasks of Knowledge graph completion, link prediction in homogenous graphs, and inductive relation prediction. The proposed method performs better than the baseline models in all the tasks.
4. In addition, authors also perform ablation study with different MESSAGE & AGGREGATION functions. Furthermore, they also analyse the performance of NBFNet on different categories of relations.

**Limitations And Societal Impact:**

Please refer to the Main Review section.

If authors include more recent baselines in their experimental comparison, I am open to revising my recommendation.

**Main Review:**

1. The proposed method seems novel, the idea of using a parameterised Bellman-Ford algorithm is a neat one.
2. AGGREGATE functions used in the paper are sum, mean, or max followed by a linear transformation and a non-linear activation, and PNA. Just curios, why didn't the authors not consider a weighted sum, with learnable weights? All the paths may not be equally important for computing the representation.
3. I am fairly satisfied with the quality and writing of the paper, I was able to follow the ideas in the paper without much struggle.
4. I am not completely satisfied with the experimental results.  NBFNet outperforms the other baselines in all the mentioned tasks, but the authors need to compare with more recent baselines.
Authors should include at least a few of the following more recent baselines in their comparison. Or discuss why they are not relevant, if that is the case.
Currently, the most recent baseline they include is from 2018.

- LowFER: Low-rank Bilinear Pooling for Link Prediction, Amin et al, ICML 2020.
- A Multi-Scale Approach for Graph Link Prediction. Cai et al, AAAI 2020.
-  Learning Hierarchy-Aware Knowledge Graph Embeddings for Link Prediction, Zhang et al , AAAI 2020.
- Contextual Parameter Generation for Knowledge Graph Link Prediction, Stocia et al, AAAI 2020.
- Link Prediction with Persistent Homology: An Interactive View, Yan et al, ICML 2021.


=====
Post author-response.
After reading all the reviews and authors response. I have decided to keep my score the same.

**Time Spent Reviewing:**

4-6 hours

---

> ### Author Response · Authors · 2021-08-10
> **Response to Reviewer 4kGi**
>
> Thanks for your comments and suggestions! Here is our response to your concerns and comments.
>
> **Q1: NBFNet outperforms the other baselines in all the mentioned tasks, but the authors need to compare with more recent baselines. Authors should include at least a few of the following more recent baselines in their comparison. Or discuss why they are not relevant, if that is the case.**
>
> A1: Thanks for providing the recent baselines. We will add those baselines in our revision to provide a more fair comparison to our readers, as long as the baselines use the same evaluation protocol as ours.
>
> Among the baselines you provided, LowFER[1] and HAKE[2] are directly comparable to our model in knowledge graph completion. CoPER[3] is only evaluated for tail prediction $p(v|u,q)$, and we provide the result of NBFNet under their evaluation protocol. TLC-GNN[4] has link prediction result on PubMed, and we can use their official code to obtain results on Cora and Citeseer. The only exception is mLink[5], which was evaluated on datasets different from ours and is not released. Here is the comparison between NBFNet and recent baselines.
>
> Results on knowledge graph completion. Ranking is computed for both head and tail entities, which is same as the evaluation setting used in the paper.
>
> | FB15k-237 | MR | MRR | HITS@1 | HITS@3 | HITS@10 |
> |-----------|-----|-------|--------|--------|---------|
> | HAKE | - | 0.346 | 0.250 | 0.381 | 0.542 |
> | LowFER | - | 0.359 | 0.266 | 0.396 | 0.544 |
> | NBFNet | 139 | **0.378** | **0.285** | **0.417** | **0.563** |
>
> | WN18RR | MR | MRR | HITS@1 | HITS@3 | HITS@10 |
> |--------|-----|-------|--------|--------|---------|
> | HAKE | - | 0.497 | 0.452 | 0.516 | 0.582 |
> | LowFER | - | 0.465 | 0.434 | 0.479 | 0.526 |
> | NBFNet | 653 | **0.547** | **0.490** | **0.573** | **0.661** |
>
> Results on knowledge graph completion. Ranking is computed for tail entities only.
>
> | FB15k-237 | MR | MRR | HITS@1 | HITS@3 | HITS@10 |
> |-----------|------|-------|--------|--------|---------|
> | CoPER | - | 0.426 | 0.322 | - | 0.629 |
> | NBFNet | 97.8 | **0.471** | **0.368** | 0.522 | **0.669** |
>
> | WN18RR | MR | MRR | HITS@1 | HITS@3 | HITS@10 |
> |--------|-----|-------|--------|--------|---------|
> | CoPER | - | 0.483 | 0.441 | - | 0.561 |
> | NBFNet | 882 | **0.560** | **0.504** | 0.584 | **0.677** |
>
> Results on homogeneous graph link prediction.
>
> | Link Prediction | Cora (AUROC) | Citeseer (AUROC) | PubMed (AUROC) |
> |-----------------|--------------|------------------|----------------|
> | TLC-GNN | 0.934 | 0.909 | 0.970 |
> | NBFNet | **0.963** | **0.956** | **0.988** |
>
> We also notice that there are some recent works like CapsE[6] and KBAT[7] achieve even better results than NBFNet on FB15k-237 dataset. However, as suggested by this paper[8], these methods have issues in the implementation of ranking in evaluation.
>
> [1] Saadullah Amin, et al. "LowFER: Low-rank bilinear pooling for link prediction." ICML, 2020.
>
> [2] Zhanqiu Zhang, et al. "Learning hierarchy-aware knowledge graph embeddings for link prediction." AAAI 2020.
>
> [3] George Stoica, et al. "Contextual parameter generation for knowledge graph link prediction." AAAI 2020.
>
> [4] Zuoyu Yan, et al. "Link Prediction with Persistent Homology: An Interactive View." ICML, 2021.
>
> [5] Lei Cai and Shuiwang Ji. "A multi-scale approach for graph link prediction." AAAI 2020.
>
> [6] Thanh Vu, et al. "A capsule network-based embedding model for knowledge graph completion and search personalization." NAACL
> 2019.
>
> [7] Deepak Nathani, et al. "Learning attention-based embeddings for relation prediction in knowledge graphs." ACL 2019.
>
> [8] Zhiqing Sun, et al. "A Re-evaluation of Knowledge Graph Completion Methods." ACL 2020.
>
> **Q2: AGGREGATE functions used in the paper are sum, mean, or max followed by a linear transformation and a non-linear activation, and PNA. Just curious, why didn't the authors not consider a weighted sum, with learnable weights? All the paths may not be equally important for computing the representation.**
>
> A2: We experiment with additionally learnable weights in the Aggregate function. The weights for messages are learned in a self-attention fashion similar to GAT[9]. The following tables compare the results of the original NBFNet and NBFNet with self attention. Self attention improves the performance on FB15k-237, but not on WN18RR. We conjecture the reason is that WN18RR has fewer relations and less average degrees than FB15k-237, which makes attention mechanism less useful.
>
> [9] Petar Veličković, et al. "Graph attention networks." ICLR 2018.
>
> Results on inductive relation prediction.
>
> | FB15k-237 | v1 | v2 | v3 | v4 |
> |--------------------|-----------|-----------|-----------|-----------|
> | NBFNet | **0.692** | 0.858 | **0.898** | 0.923 |
> | NBFNet + attention | **0.690** | **0.906** | 0.865 | **0.941** |
>
> | WN18RR | v1 | v2 | v3 | v4 |
> |--------------------|-----------|-----------|-----------|-----------|
> | NBFNet | **0.942** | **0.895** | **0.900** | **0.881** |
> | NBFNet + attention | **0.944** | **0.893** | 0.889 | 0.874 |

---

> ### Author Response · Authors · 2021-08-15
> **Further suggestions are welcome**
>
> Thanks for recognizing our work. Your suggestion on recent baselines helped us to shape this paper towards a fairer position in the link prediction community. Besides the baselines, if you have any other question or suggestion regarding the paper, feel free to tell us and we will provide more feedbacks.

---

### Official Review · Reviewer_5weJ · 2021-07-18

**Rating:** 6
**Confidence:** 3

**Summary:**

Summary: In this study the authors propose a general framework to learn neural INDICATOR, MESSAGE and AGGREGATE functions for the generalized Bellman-Ford (BF) algorithm. They apply the neural BF to link prediction problems in multirelational knowledge graphs in the in-sample setting as well as in the out-of-sample setting, and in homogeneous graphs. On a practical level the approach trades compute with memory and shows good results on benchmark datasets.

**Limitations And Societal Impact:**

Limitations and societal impact have been addressed.

**Main Review:**

Originality: The approach combines the generalized BF algorithm with learnables neural operators and sets the approach well in context with previous work.

Quality: The paper does not provide a proof or error bounds on the solution it can find. The study describes the proposed approach formally well but its usefulness is mostly backed up by empirical results. The approach is explored well, however, an aspect that was missing ---as the approach is very expensive--- it would be fair and important to show the compute trade-off as well as the memory trade-off for each of the groups (path-based, embedding, GNN) with the proposed approach in the performance comparison, as I don’t think it that they should be decoupled in such comparisons. While it does well regarding memory with embedding based methods, it does a lot worse compared to GRCN and even more so compared to embedding based methods.

The experimental results do not cite state of the art results for FB15k-237 and WNRR which are for FB15k-237 much closer to the results obtained by the proposed approach. Complex achieves 0.37 MRR on Fb15k-237 and 0.49 MRR on WN18RR [1]. There are methods and dataset for out-of-sample settings for knowledge graph completion [2] that should be considered in the comparison.

Another concern about the knowledge graph completion experiment ---and the evaluation description was not explicit about this and no code was provided--- is, which graph is known by the model during evaluation? Does it consist of (1) the training triples, or (2) the training-triples + the validation triples, or (3) the training+validation+testing triples (minus each testing triple once it is under evaluation of course)? Sometimes path-based methods use (2) or (3) which naturally leads to better results. Can the authors explicitly confirm which of the settings was the one that was used? For knowledge graph embedding methods the known graph corresponds to (1), i.e. only the training triples. If this is different in this study, then a performance comparison between embedding methods and this proposed method is not possible.

One attractive property of the model is that the highest scoring paths can be inspected which can lead to new insights about datasets.

Clarity: The paper is written well and the presentation is great. The experimental setup could be clearer.

Significance: If the concerns can be ruled out, then the result would be important.  The method achieves good numbers on benchmark datasets. There is great interest in interpretable methods as well, which this method also does provide.


**Time Spent Reviewing:**

2

---

> ### Author Response · Authors · 2021-08-10
> **Response to Reviewer 5weJ**
>
> Thanks for your constructive comments and suggestions! Here is our response to your concerns.
>
> **Q1: The paper does not provide a proof or error bounds on the solution it can find.**
>
> A1: Yes, you are right. But note that the focus of this paper is to develop an effective and efficient algorithm for link prediction on both knowledge graphs and homogeneous graphs. We leave the theoretical analysis of our algorithm as the future work.
>
> **Q2: An aspect that was missing ---as the approach is very expensive--- it would be fair and important to show the compute trade-off as well as the memory trade-off for each of the groups (path-based, embedding, GNN) with the proposed approach in the performance comparison, as I don’t think it that they should be decoupled in such comparisons. While it does well regarding memory with embedding based methods, it does a lot worse compared to RGCN and even more so compared to embedding based methods.**
>
> A2: We agree that NBFNet is more expensive than prevalent embedding methods (TransE, DistMult, etc.) and RGCN. However, their good efficiency is at the expense of being unable to handle the inductive setting. This is because these methods all introduce an embedding for each entity, allowing them to memorize a feature for each entity and hence achieve better efficiency. We will clarify that NBFNet is scalable among inductive methods, which is supported by the complexity proofs in Appendix C.
>
> **Q3: Which graph is known by the model during evaluation? Does it consist of (1) the training triples, or (2) the training-triples + the validation triples, or (3) the training+validation+testing triples (minus each testing triple once it is under evaluation of course)?**
>
> A3: We only use the training triplets as the input graph structure during test, as setting (1) in your question. It is also same as the setting for all baselines in the paper. We will release the code to let others verify that and reproduce NBFNet once the paper is accepted.
>
> **Q4: Complex achieves 0.37 MRR on Fb15k-237 and 0.49 MRR on WN18RR [1]. There are methods and datasets for out-of-sample settings for knowledge graph completion [2] that should be considered in the comparison.**
>
> A4: Sorry we can't find the references you mentioned. If you can provide the paper names and if they share the same evaluation setting with ours, we will add them to the paper. We have already added the baselines suggested by Reviewer 4kGi.

---

> > ### Comment · Reviewer_5weJ · 2021-08-10
> > **Missing references**
> >
> > Here are the missing references from the review.
> >
> > [1] Lacroix et al. Canonical tensor decomposition for knowledge base completion.
> >
> > [2] Albooyeh et al. Out-of-sample representation learning for knowledge graphs.

---

> > > ### Author Response · Authors · 2021-08-11
> > > **Response to Reviewer 5weJ**
> > >
> > > Thanks for your quick reply.
> > >
> > > We read through the papers you provided. ComplEx-N3[1] shares the same setting as NBFNet and we will add ComplEx-N3 in the comparison of knowledge graph completion methods. For the out-of-sample paper[2], it studies the setting where test entities are unobserved but connected to the training entities. In our paper, we study the inductive setting introduced by [3] where training and test knowledge graphs are disjoint, and only share the vocabulary of relations. Therefore, methods in [2] can't be applied to the inductive setting.
> > >
> > > Results on knowledge graph completion.
> > >
> > > | FB15k-237 | MR | MRR | HITS@1 | HITS@3 | HITS@10 |
> > > |------------|-----|------|--------|--------|---------|
> > > | ComplEx-N3 | - | 0.37 | - | - | **0.56**|
> > > | NBFNet | 139 | **0.378** | 0.285 | 0.417 | **0.563** |
> > >
> > > | WN18RR | MR | MRR | HITS@1 | HITS@3 | HITS@10 |
> > > |------------|-----|------|--------|--------|---------|
> > > | ComplEx-N3 | - | 0.48 | - | - | 0.57 |
> > > | NBFNet | 653 | **0.547** | 0.490 | 0.573 | **0.661** |
> > >
> > > [1] Timothée Lacroix et al. "Canonical tensor decomposition for knowledge base completion." ICML 2018.
> > >
> > > [2] Marjan Albooyeh et al. "Out-of-sample representation learning for knowledge graphs." EMNLP 2020.
> > >
> > > [3] Komal Teru et al. "Inductive relation prediction by subgraph reasoning." ICML 2020.

---

> ### Author Response · Authors · 2021-08-15
> **Further suggestions are welcome**
>
> Thanks for your attention and response to our feedback. We are not sure if our feedback has addressed your concerns, especially on the questions of error bounds and comparison to RGCN / embedding methods. We are happy to launch more discussions, and receive further suggestions to improve our paper. Please feel free to reach out if you have any question.

---

> > ### Comment · Reviewer_5weJ · 2021-08-19
> > **Thank you for your response**
> >
> > Your comments have addressed my questions sufficiently. At this point I have no further comments or questions.

---

### Author Response · Authors · 2021-08-10
**Summary of Responses**

We would like to thank all reviewers for your time and constructive comments. Here is a brief summary of important questions from all reviewers.

**Writing & Clarification**

1. **Scalability (Reviewer 5weJ, QoNE):** We clarify that NBFNet is scalable among inductive methods. NBFNet can scale to the knowledge graph in OGB-LSC (87M entities, 504M triplets) with a simple neighborhood sampling technique.
2. **Inductive setting (Reviewer QoNE):** We clarify that inductive relation prediction requires models to generalize to new knowledge graphs **without node features**, which can't be solved by RGCN or simple inductive baselines over node features.
3. **Experiment setting (Reviewer 5weJ, QoNE):** We select our model based on the best results on the validation set. For transductive settings, we only use the training graph as the input in evaluation, which is the same as the setting for baselines in our paper.
4. **Relation to ID-GNN (Reviewer HJVx):** We agree the instantiation of NBFNet shares some spirits with ID-GNN and add a citation to ID-GNN. However, NBFNet is fundamentally different from ID-GNN in the tasks, motivations and techniques.
5. **Release (Reviewer 5weJ, QoNE):** We will release the code later for reproduction.

**Experiments**

1. **Recent baselines (Reviewer 5weJ, 4kGi):** We add the recent baselines provided by the reviewers to present a more fair comparison to the readers.
2. **Learnable weights in Aggregate function (Reviewer 4kGi):** We add an experiment that uses self attention to learn the weight for each message. It improves the performance on FB15k-237 but not on WN18RR.
3. **OGB & KDDCup 2021 (Reviewer HJVx):** We add an experiment on ogbl-biokg dataset. NBFNet achieves the 2nd best performance compared to the results reported on the official leaderboard. NBFNet was also submitted to the OGB-LSC challenge, and achieved 12th rank of 28 teams. The single model result of NBFNet is better than the best single model results reported by the winner and the runner up.

---

### Decision · Program_Chairs · 2021-09-27

**Decision:**

Accept (Poster)

**Comment:**

This paper proposed NBFNet a framework based on GNNs for representation learning tasks on graphs. All the reviewers agreed that the paper presents quite impressive empirical results and the proposed method is novel. There was also a general agreement amongst the reviewers that the paper seemed to be written in a rush and hence clarity was an issue. After taking into account the authors' responses it was decided that the paper is marginally above the acceptance threshold. I recommend acceptance and strongly urge the authors to improve the final camera ready version and take the reviewers' comments into account.